# GluA4 facilitates cerebellar expansion coding and enables associative memory formation

**Katarzyna Kita[1,2], Catarina Albergaria[3], Ana S Machado[3], Megan R Carey[3], Martin Müller[1,2], Igor Delvendahl[1,2]***

[1]Department of Molecular Life Sciences, University of Zurich, Zurich, Switzerland; [2]Neuroscience Center Zurich, Zurich, Switzerland; [3]Champalimaud Neuroscience Programme, Champalimaud Centre for the Unknown, Lisbon, Portugal

**Abstract** AMPA receptors (AMPARs) mediate excitatory neurotransmission in the central nervous system (CNS) and their subunit composition determines synaptic efficacy. Whereas AMPAR subunits GluA1–GluA3 have been linked to particular forms of synaptic plasticity and learning, the functional role of GluA4 remains elusive. Here, we demonstrate a crucial function of GluA4 for synaptic excitation and associative memory formation in the cerebellum. Notably, GluA4-knockout mice had ~80% reduced mossy fiber to granule cell synaptic transmission. The fidelity of granule cell spike output was markedly decreased despite attenuated tonic inhibition and increased NMDA receptor-mediated transmission. Computational network modeling incorporating these changes revealed that deletion of GluA4 impairs granule cell expansion coding, which is important for pattern separation and associative learning. On a behavioral level, while locomotor coordination was generally spared, GluA4-knockout mice failed to form associative memories during delay eyeblink conditioning. These results demonstrate an essential role for GluA4-containing AMPARs in cerebellar information processing and associative learning.

*For correspondence:
igor.delvendahl@uzh.ch

Competing interest: See
page 22

Reviewing editor: Solange P
Brown, Johns Hopkins University,
United States

## Introduction

AMPA receptors (AMPARs) are essential for excitatory neurotransmission in the central nervous system (CNS). AMPARs are tetramers assembled from a combination of four subunits, GluA1–GluA4, that have distinct properties (*Traynelis et al., 2010*). AMPAR subunit expression shows strong regional variations (*Geiger et al., 1995*; *Schwenk et al., 2014*; *Sjöstedt et al., 2020*), suggesting specific functions in different types of neurons. Previous studies have revealed roles for AMPAR subunits GluA1–GluA3 in diverse forms of synaptic plasticity across the CNS (*Citri et al., 2010*; *Gutierrez-Castellanos et al., 2017*; *Renner et al., 2017*; *Roth et al., 2020*; *Shi et al., 2001*; *Silva et al., 2019*; *Steinberg et al., 2006*; *Zamanillo et al., 1999*). By contrast, much less is known about the function of the GluA4 subunit.

The GluA4 subunit confers rapid kinetics and large conductance to AMPARs (*Mosbacher et al., 1994*; *Swanson et al., 1997*). Yet, its expression is confined to few types of neurons in the adult brain (*Keinänen et al., 1990*; *Monyer et al., 1991*). Deletion of GluA4 impairs excitatory input to certain neurons in the auditory brainstem (*Yang et al., 2011*), the thalamus (*Paz et al., 2011*; *Seol and Kuner, 2015*), and the hippocampus (*Fuchs et al., 2007*). In the cerebellum, granule cells (GCs)—the most abundant neurons in the mammalian brain—heavily express GluA4 (*Cathala et al., 2005*; *Hollmann and Heinemann, 1994*; *Mosbacher et al., 1994*; *Schwenk et al., 2014*), but the significance of GluA4 for GCs and cerebellar circuit function remains unclear.

The cerebellum plays an important role in sensorimotor integration, motor coordination, motor learning and timing, as well as cognition (*Diedrichsen et al., 2019*; *Raymond and Medina, 2018*).

The underlying computations are very fast, allowing millisecond precision in the calibration and timing of movement (*Heck et al., 2013*; *Osborne et al., 2007*). This astonishing speed is achieved by the cerebellar cortex with a highly conserved network structure: Sensory and motor information enter the cerebellar cortex via mossy fibers (MFs) that contact a large number of GCs. These small, numerous neurons greatly outnumber MFs, thus providing a large expansion of coding space (*Albus, 1971*; *Marr, 1969*). GCs relay information via their parallel fibers (PFs) to Purkinje cells (PCs), the sole output neurons of the cerebellar cortex. Accumulating evidence suggests that the MF→GC synapse has evolved mechanisms allowing for transmission with exceptionally high rates and precision (*Delvendahl and Hallermann, 2016*; *DiGregorio et al., 2002*; *Rancz et al., 2007*). However, little is known about the molecular framework and functional role of the rapid signal integration in cerebellar GCs.

Here, we used electrophysiological recordings, computational modeling, and behavioral analyses to study cerebellar function in adult GluA4-knockout (GluA4-KO) mice. Deletion of GluA4 resulted in selective impairment of MF→GC transmission. Despite compensatory changes in tonic inhibition and NMDA receptor (NMDAR)-mediated synaptic input, the pronounced decrease of AMPAR-mediated excitation caused a severe deficit in synaptic integration during high-frequency transmission. These synaptic changes impaired pattern separation and learning performance of a feedforward network model. On a behavioral level, GluA4-KO mice displayed normal locomotor coordination, but a complete absence of eyeblink conditioning. Our findings demonstrate the importance of the GluA4 AMPAR subunit for cerebellar input layer synaptic function and associative memory formation.

## Results

### Selective impairment of cerebellar MF→GC synapses in GluA4-KO mice

GluA4 shows a regional expression pattern in the CNS with the strongest detection in the cerebellum (*Sjöstedt et al., 2020*). Within the cerebellar cortex, this AMPAR subunit has been described in Bergmann glia (*Saab et al., 2012*) and in GCs (*Mosbacher et al., 1994*), where GluA4 is the major AMPAR subunit at MF→GC synapses (*Delvendahl et al., 2019*). To investigate the role of GluA4 for cerebellar function, we performed whole-cell patch-clamp recordings at excitatory synapses in slices of adult wild-type (WT) and GluA4-KO mice (*Figure 1A*). Recordings were made in the anterior vermis that mainly receives sensory MF input (lobules III–VI; *Giovannucci et al., 2017*; *Witter and De Zeeuw, 2015*). GluA4-KO GCs displayed strongly diminished synaptic responses upon MF stimulation, with an ~80% reduction in excitatory postsynaptic currents (EPSCs; 13.4 ± 1.1 pA vs. 76.2 ± 7.5 pA; p < 0.0001; *Figure 1B–C*; *Delvendahl et al., 2019*). Consistent with the fast kinetics of GluA4-containing AMPARs (*Mosbacher et al., 1994*), the EPSC decay was slower in GluA4-KO GCs than in WT (3.1 ± 0.3 ms vs. 1.8 ± 0.2 ms; p < 0.001; *Figure 1C*). We next asked if GluA4 is involved in transmission at other synapses of the cerebellar input layer. Golgi cells (GoCs) provide feedforward and feedback inhibition to GCs (*Duguid et al., 2015*), but it remains unknown if GluA4-containing AMPARs play a role in GoC excitation. At MF→GoC connections (*Figure 1D*), EPSC amplitudes and decay kinetics were comparable between WT and GluA4-KO (*Figure 1E–F*), suggesting that GluA4 does not contribute to MF→GoC transmission. The strong reduction of excitatory input onto GCs in GluA4-KO animals might alter GC output via their PFs. We therefore investigated PF inputs to GoCs (PF→GoC synapses; *Figure 1G*), which were similar between WT and GluA4-KO mice (*Figure 1H–I*). Consistent with the similar EPSC amplitudes, spontaneous EPSCs and paired-pulse ratios were also not altered in GoCs of GluA4-KO mice (*Figure 1—figure supplement 1*). Whereas PCs do not express GluA4 (*Lambolez et al., 1992*), loss of this AMPAR subunit in Bergmann glial cells might impact PF→PC transmission (*Saab et al., 2012*). To examine PF→PC synapses (*Figure 1J*), we stimulated in the molecular layer with increasing intensity and recorded PF→PC EPSCs in WT and GluA4-KO PCs (*Figure 1K*). The slope of the stimulus-response relationship was comparable between genotypes (*Figure 1L*; *Figure 1—figure supplement 2*), which indicates normal recruitment of PFs. Likewise, EPSC decay kinetics (*Figure 1L*) as well as spontaneous EPSC amplitudes and paired-pulse ratios were similar between WT and KO PCs (*Figure 1—figure supplement 2*), suggesting that GluA4 does not contribute directly to PF→PC synaptic transmission. The recordings from PF→GoC and PF→PC synapses thus indicate that PF output is functionally intact in GluA4-KO mice. Moreover, gross cerebellar organization was unaltered in GluA4-KO mice (*Figure 1—figure supplement 3*).

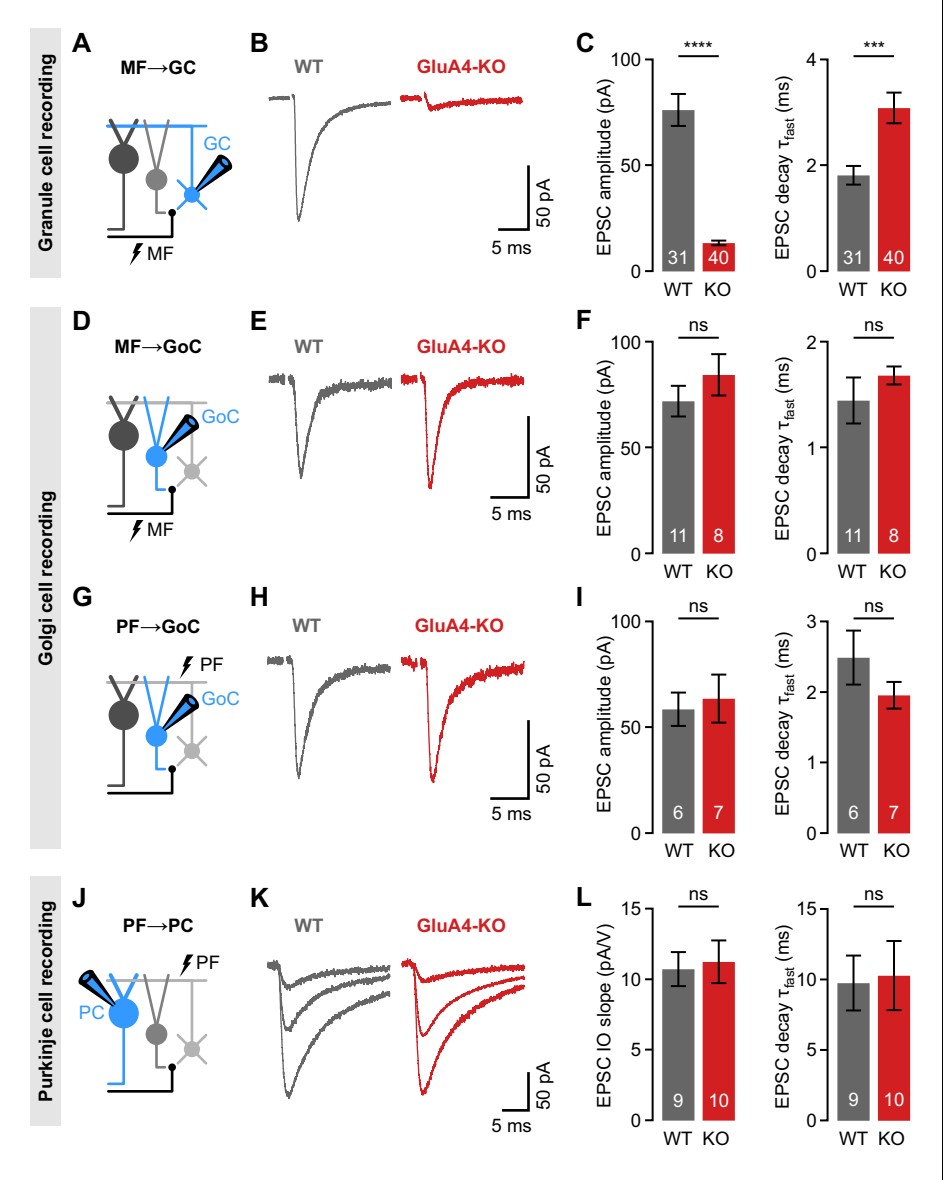

**Figure 1.** Selective impairment of cerebellar mossy fiber (MF)→granule cell (GC) synapses in GluA4-knockout (GluA4-KO) mice. (**A**) Schematic of recordings from MF→GC connections. (**B**) Example excitatory postsynaptic currents (EPSCs) recorded from wild-type (WT) and KO GCs by single MF stimulation. (**C**) Average EPSC amplitude (left; p < 0.0001; Cohen's d = −2.23) and average EPSC fast decay time constant (right; p < 0.001; d = 0.84) for WT and KO MF→GC EPSCs. Data are redrawn from *Delvendahl et al., 2019*. (**D**) Recordings from MF→Golgi cell (GoC) connections. (**E**) Example EPSCs upon MF stimulation recorded from GoCs. (**F**) Average EPSC amplitude (left; p = 0.32; d = 0.49) and fast decay time constant (right; p = 0.33; d = 0.42) for WT and KO MF→GoC EPSCs. (**G**) Recordings from parallel fiber (PF)→GoC connections. (**H**) Example EPSCs recorded from GoCs upon PF stimulation. (**I**) Average EPSC amplitude (left; p = 0.73; d = 0.19) and fast decay time constant (right; p = 0.25; d = −0.73) for WT and KO PF→GoC EPSCs. (**J**) Recordings from PF→Purkinje cell (PC) connections. (**K**) Example EPSCs recorded from PCs by stimulating afferent PFs with increasing stimulation strength (displayed stimulation intensities: 7, 12, and 17 V). (**L**) Average linear slope of the stimulus-response relationship (left; p = 0.79; d = 0.12) and fast decay time constant (right; p = 0.87; d = 0.08) for WT and KO PF→PC EPSCs. Data are means ± SEM.

The online version of this article includes the following source data and figure supplement(s) for figure 1:

**Source data 1.** Numerical data plotted in *Figure 1*.

**Figure supplement 1.** Unaltered transmission at excitatory synapses in cerebellar Golgi cells (GoCs).

**Figure supplement 2.** Unaltered transmission at excitatory synapses in cerebellar Purkinje cells (PCs).

*Figure 1 continued on next page*

*Figure 1 continued*

**Figure supplement 3.** Unaltered gross cerebellar organization.

Together, our findings demonstrate that GluA4 is indispensable for fast neurotransmission at MF→GC synapses and suggest a GC-specific impairment of excitatory transmission at the cerebellar input layer in GluA4-KO mice.

## Reduced tonic inhibition increases the GC input-output relationship

Loss of GluA4 severely compromises synaptic excitation of cerebellar GCs. Changes in both excitatory and inhibitory synaptic input may influence a neuron's excitability (*Aizenman and Linden, 2000*; *Desai et al., 1999*; *Karmarkar and Buonomano, 2006*). To investigate if GluA4-KO alters the input-output relationship of GCs, we quantified GC spiking upon incrementing tonic current injections (*Figure 2A-B*). Under control conditions, GluA4-KO GCs showed higher firing frequencies than WT, reflecting increased excitability (linear mixed effects analysis: current × genotype interaction, $\chi^2(1) = 35.48$, $p < 0.001$; *Figure 2B–C*). Because GoC-mediated tonic inhibition influences the input-output relationship of GCs both in slices (*Brickley et al., 2001*; *Hamann et al., 2002*; *Mitchell and Silver, 2003*; *Rothman et al., 2009*) and in vivo (*Chadderton et al., 2004*), we probed GC firing in the presence of the GABA$_A$ receptor (GABA$_A$R) blocker bicuculline. Interestingly, the input-output relationship was similar between KO and WT GCs in the presence of bicuculline (linear mixed effects analysis: current × genotype interaction, $\chi^2(1) = 0.61$, $p = 0.43$; *Figure 2B–C*). To further assess the influence of inhibition on GC excitability, we analyzed gain and rheobase of the input-output relationship. Blocking inhibition increased the gain and reduced the rheobase current in WT (*Figure 2D*), consistent with previous studies (*Hamann et al., 2002*; *Rothman et al., 2009*; *Rudolph et al., 2020*). By contrast, bicuculline had little effect on gain and rheobase in GluA4-KO GCs (ANOVA gain: genotype × drug: $F = 4.24$, $p = 0.041$, $\eta^2_p = 0.028$; rheobase: genotype × drug: $F = 2.00$, $p = 0.16$, $\eta^2_p = 0.014$; *Figure 2D*), indicating that tonic inhibition is reduced in these mice.

Persistent activation of δ- and α6-containing extrasynaptic GABA$_A$Rs causes a tonic inhibitory conductance in cerebellar GCs (*Brickley et al., 2001*; *Hamann et al., 2002*; *Stell et al., 2003*) that also influences their input-output relationship (*Figure 2—figure supplement 1*; *Mitchell and Silver, 2003*; *Rudolph et al., 2020*). We directly investigated tonic inhibition by isolating inhibitory conductance and bath-applying bicuculline. Indeed, GluA4-KO GCs had a smaller root-mean-square noise of the baseline holding current and smaller bicuculline-sensitive conductance compared with WT ($39.0 \pm 9.6$ pS/pF vs. $83.0 \pm 13.3$ pS/pF, $p = 0.022$; *Figure 2E–F*). These findings show that tonic GABA$_A$R-mediated inhibition is reduced in GluA4-KO GCs, in line with the lack of effect of bicuculline on the input-output relationship in these animals. The GABA$_A$R α6-subunit mediates tonic inhibition of GCs (*Brickley et al., 2001*). Interestingly, Western blot analyses revealed slightly lower α6 GABA$_A$R levels in GluA4-KO cerebella (*Figure 2—figure supplement 2*), consistent with the reduction of tonic inhibition in GCs. By contrast, the amplitudes and frequencies of spontaneous inhibitory postsynaptic currents (sIPSCs) were not different between WT and GluA4-KO GCs (*Figure 2—figure supplement 2*), indicating that phasic inhibition is not altered. To verify the specificity of our findings, we also studied the input-output relationship in GCs of heterozygous GluA4 (GluA4-HET) mice, which have reduced GluA4 levels (*Figure 2—figure supplement 3*) but no change in MF→GC EPSC amplitudes (*Figure 3—figure supplement 1*; *Delvendahl et al., 2019*). We did not observe indications of altered inhibition in GluA4-HET GCs (*Figure 2—figure supplement 3*). Thus, the strongly impaired excitatory synaptic input to GluA4-KO GCs is accompanied by a modulation of tonic inhibition, resulting in an increased input-output relationship.

## Shunting inhibition controls excitatory postsynaptic potential size in cerebellar GCs

Tonic GABA$_A$R activation leads to shunting inhibition of GCs that may affect their responses to excitatory input (*Brickley et al., 2001*; *Mitchell and Silver, 2003*). The reduction of tonic inhibition in GluA4-KO GCs is therefore expected to impact excitatory postsynaptic potentials (EPSPs). To assess the effect of shunting inhibition in WT and GluA4-KO GCs, we recorded MF→GC EPSCs and EPSPs in blocker-free solution ('control', *Figure 3A*). As expected, EPSC amplitudes were strongly

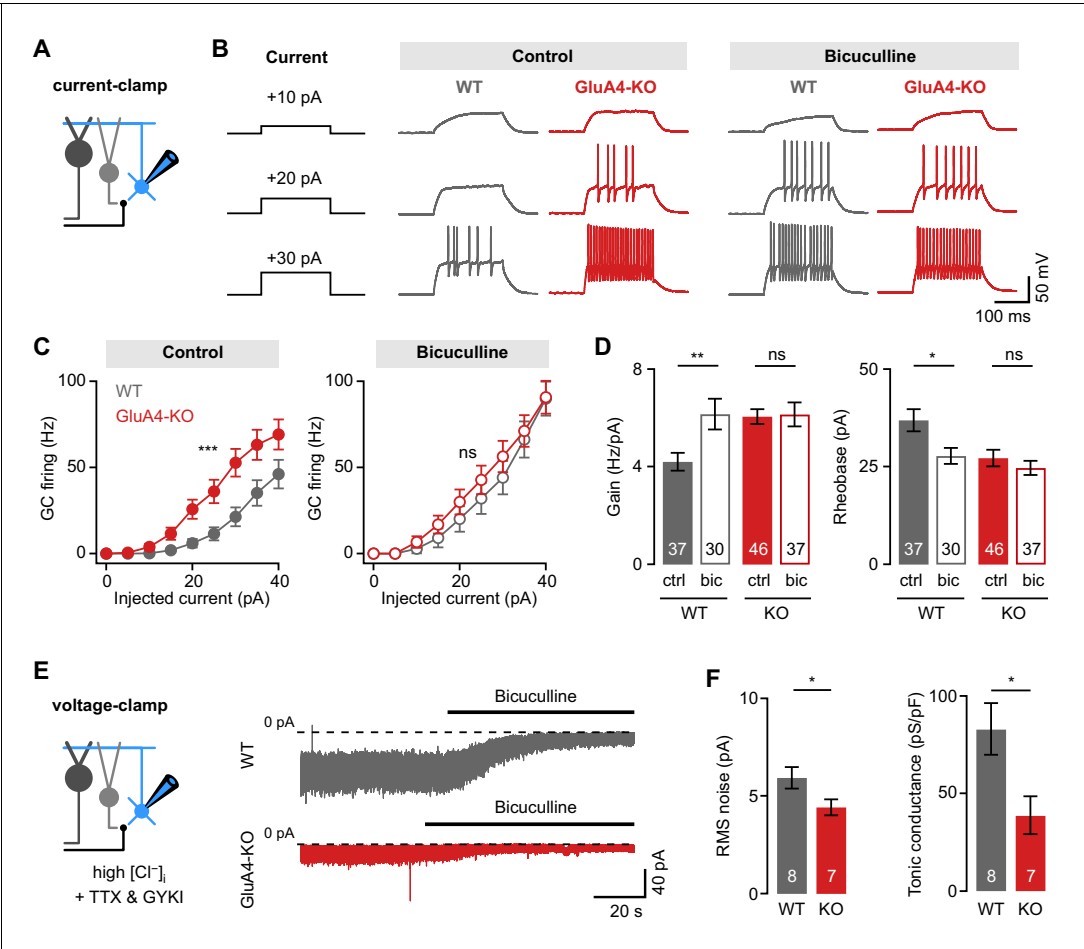

**Figure 2.** Reduced tonic inhibition increases the granule cell (GC) input-output relationship. (**A**) Current-clamp recordings from GCs. (**B**) Example responses to somatic current injection of indicated amplitude for control and in the presence of 10 µM bicuculline. (**C**) Average GC firing frequency plotted vs. injected current for wild type (WT) and knockout (KO) in control and bicuculline. (**D**) Average gain and rheobase, calculated from the frequency-current curves in (**C**). In the presence of bicuculline, KO GCs had gain and rheobase values comparable to WT. (**E**) Recording of tonic holding current. GABAergic currents were isolated using 20 µM GYKI-53655 and an intracellular solution with high [Cl⁻] to maximize GABA$_A$ receptor-mediated currents (see Materials and methods). Right: Example recordings of bicuculline wash-in (10 µM) for WT and KO GCs. (**F**) Average root-mean-square noise before wash-in of bicuculline (left; p = 0.048; d = −1.11) and average tonic conductance density (right; p = 0.020; d = −1.35) for WT and GluA4-KO. Tonic conductance was calculated from the bicuculline-sensitive current. Data are means ± SEM.

The online version of this article includes the following source data and figure supplement(s) for figure 2:

**Source data 1.** Numerical data plotted in *Figure 2*.

**Figure supplement 1.** Tonic inhibition in a cerebellar granule cell (GC) integrate-and-fire model.

**Figure supplement 2.** Phasic inhibition in cerebellar granule cells (GCs) of wild-type (WT) and GluA4-knockout (GluA4-KO) mice.

**Figure supplement 3.** Normal tonic and phasic inhibition in heterozygous GluA4 (GluA4-HET) granule cells (GCs).

diminished in GluA4-KO GCs, as were EPSP amplitudes (*Figure 3B–C*). Intriguingly, the amplitude reduction relative to WT was smaller for EPSPs than for EPSCs (69.4 ± 2.1% vs. 80.6 ± 1%, p < 0.001, d = −1.02; *Figure 3C*). To test if reduced tonic inhibition underlies this effect, we recorded EPSPs in the presence of bicuculline, which increased EPSP amplitudes in WT, but not in GluA4-KO GCs (ANOVA drug × genotype: F = 8.00, p = 0.005, $\eta^2_p$ = 0.056; *Figure 3D–E*). We also observed that bicuculline enhanced EPSP amplitudes during high-frequency train stimulation in WT, but not in GluA4-KO GCs (ANOVA drug × genotype: F = 48.77, p < 0.001, $\eta^2_p$ = 0.19; *Figure 3F–G*). In contrast, bicuculline had no differential effect on EPSC amplitudes (ANOVA drug × genotype: F = 0.35, p = 0.56, $\eta^2_p$ = 0.001; *Figure 3—figure supplement 2*). These results demonstrate that shunting inhibition reduces postsynaptic depolarization in response to excitatory input in WT, but not in

GluA4-KO GCs. We conclude that the reduced tonic inhibition upon loss of the major AMPAR subunit at MF→GC synapses augments synaptic depolarization in GluA4-KO GCs.

## Deletion of GluA4 impairs GC spike fidelity and precision during high-frequency transmission

How do the changes in excitatory input and input-output relationship influence GC firing in response to MF input? To address this question, we performed current-clamp recordings from GCs in combination with high-frequency MF→GC stimulation (*Figure 4A*) at a membrane potential of −70 mV, corresponding to the average resting membrane potential of GCs in vivo (*Chadderton et al., 2004*; *Powell et al., 2015*). WT GCs showed increasing firing frequencies upon 100–300 Hz stimulation (*Figure 4B–C*, *Figure 4—figure supplement 1*). We also observed MF stimulation-evoked spikes in GluA4-KO GCs, albeit at reduced frequency (ANOVA genotype: F = 23.48; p < 0.0001, $\eta^2_p$ = 0.15; *Figure 4C*) and in fewer cells (*Figure 4D*). The GC spikes elicited by high-frequency MF→GC stimulation were not only reduced in number, but also occurred with a longer delay compared to WT (ANOVA genotype: F = 26.26; p < 0.0001, $\eta^2_p$ = 0.29; *Figure 4E-F*). GluA4-KO GCs hence need to summate more EPSPs to reach spike threshold (*Figure 4—figure supplement 2*), leading to reduced reliability and impaired temporal precision of GC spikes. In contrast, spike output was not altered in GCs of GluA4-HET mice (*Figure 4—figure supplement 3*). Together, these data show a strong reduction of spiking frequency and temporal precision in GluA4-KO GCs upon high-frequency

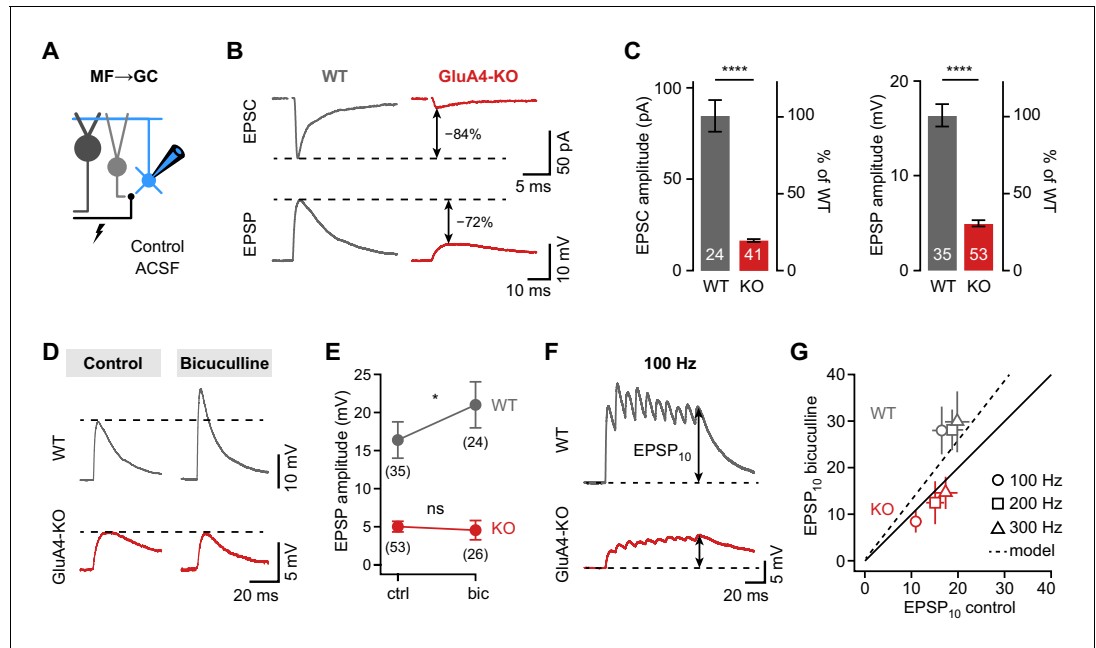

**Figure 3.** Shunting inhibition controls excitatory postsynaptic potential (EPSP) size in cerebellar granule cells (GCs). (A) Recordings from mossy fiber (MF)→GC connections in voltage- or current-clamp with blocker-free artificial cerebrospinal fluid (ACSF) ('control'). (B) Example excitatory postsynaptic current (EPSC) and excitatory postsynaptic potential (EPSP) recordings from the same cell in wild type (WT) and knockout (KO). Arrows indicate reduction compared to WT. (C) Average EPSC (left, p < 0.0001, d = −2.62) and EPSP amplitudes (right, p < 0.0001, d = −2.37) for WT and KO. Note the difference in relative reduction of EPSPs compared to EPSCs. (D) Example EPSP recordings for control and in the presence of 10 µM bicuculline. (E) Average EPSP amplitude for WT and KO. Error bars represent 95% CI. (F) Example MF→GC EPSP trains (10 stimuli, 100 Hz). (G) Average EPSP₁₀ in the presence of bicuculline vs. control for the indicated stimulation frequencies. Black line represents unity and dashed line the prediction of an integrate-and-fire model with or without tonic inhibition (*Figure 2—figure supplement 1*). Data in (C) and (G) are means ± SEM.

The online version of this article includes the following source data and figure supplement(s) for figure 3:

**Source data 1.** Numerical data plotted in *Figure 3*.

**Figure supplement 1.** Normal mossy fiber (MF)→granule cell (GC) excitatory postsynaptic currents (EPSCs) and excitatory postsynaptic potentials (EPSPs) in heterozygous GluA4 (GluA4-HET) GCs.

**Figure supplement 2.** Similar mossy fiber (MF)→granule cell (GC) excitatory postsynaptic current (EPSC) amplitudes between control and bicuculline.

synaptic input, and establish that GluA4-containing AMPARs control the timing and reliability of EPSP-spike coupling at the cerebellar input layer.

## NMDARs and reduced inhibition support spiking in GluA4-KO GCs

Given the strong reduction of MF→GC EPSCs (*Figure 1*), the occurrence of spikes upon repetitive stimulation in more than 30% of GluA4-KO GCs is surprising. We previously provided evidence for enhanced presynaptic glutamate release from GluA4-KO MF boutons (*Delvendahl et al., 2019*), which might increase the non-AMPAR component of MF→GC transmission. Indeed, isolated NMDAR-mediated MF→GC EPSCs were larger in GluA4-KO than in WT GCs (37.2 ± 2.1 pA vs. 27.5 ± 2.1 pA; p = 0.0026; d = 1.04; *Figure 5A–B*). Whereas NMDAR-EPSC paired-pulse ratios were similar between GluA4-KO and WT MF→GC synapses, statistical moments analysis of quanta (*Holler et al., 2021*) indicated an increase in presynaptic release sites without apparent changes in release probability or quantal size in GluA4-KO (*Figure 5—figure supplement 1*). Since NMDAR activation particularly supports GC excitation during repetitive input (*D'Angelo et al., 1995*), we recorded isolated AMPAR- and NMDAR-EPSCs upon high-frequency train stimulation (100 Hz, 20 stimuli). As expected, AMPAR-EPSCs were strongly decreased in GluA4-KO GCs (*Figure 5C*), leading to a pronounced reduction of cumulative EPSC charge during the train (0.9 ± 0.1 pC vs. 3.4 ± 0.2 pC; p < 0.0001; d = −3.05; *Figure 5D*). By contrast, NMDAR-EPSCs and cumulative EPSC charge were increased in GluA4-KO GCs (8.7 ± 0.9 pC vs. 5.6 ± 0.4 pC; p = 0.0063; d = 0.92; *Figure 5D*).

To investigate if reduced tonic inhibition and enlarged NMDAR-EPSCs support spiking in GluA4-KO GCs, we employed computational modeling. We fit the recorded MF→GC AMPAR and NMDAR conductance trains (G_AMPA and G_NMDA, respectively; *Figure 5E*) to model synaptic input of

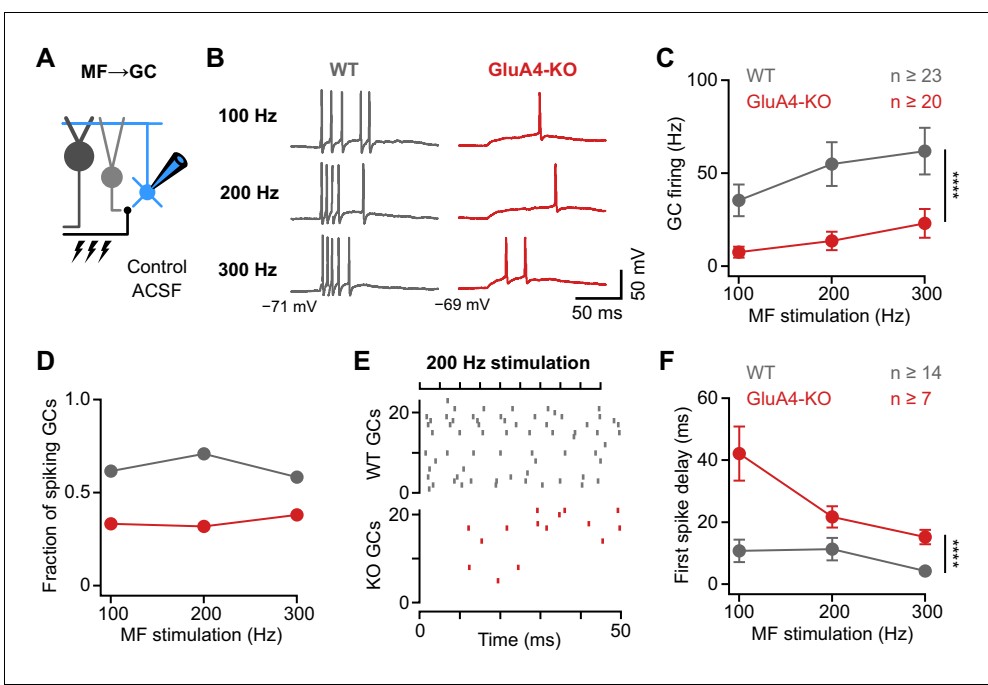

**Figure 4.** Deletion of GluA4 impairs granule cell (GC) spike fidelity and precision during high-frequency transmission. (**A**) Recordings from mossy fiber (MF)→GC connections in current-clamp. GCs were held at −70 mV. (**B**) Example voltage recordings from a wild type (WT) and GluA4-knockout (GluA4-KO) GC upon MF stimulation with 10 stimuli at increasing frequencies (indicated). (**C**) Average GC firing frequency upon MF stimulation with 100–300 Hz frequency. (**D**) Fraction of GCs showing spikes upon 100–300 Hz MF stimulation. (**E**) Spike raster plots for 200 Hz MF stimulation in WT and GluA4-KO. (**F**) Average first spike delay calculated from stimulation onset plotted vs. MF stimulation frequency. Data are means ± SEM.

The online version of this article includes the following source data and figure supplement(s) for figure 4:

**Source data 1.** Numerical data plotted in *Figure 4*.
**Figure supplement 1.** Impaired granule cell (GC) spike output in GluA4-knockout (GluA4-KO) GCs at −80 mV.
**Figure supplement 2.** Impaired granule cell (GC) spiking in an integrate-and-fire (IAF) GC model.
**Figure supplement 3.** Normal granule cell (GC) spike output in heterozygous GluA4 (GluA4-HET) GCs.

an integrate-and-fire neuron (*Rothman et al., 2009*). Simulations reproduced the experimentally observed GC firing and first spike delay well for WT and GluA4-KO upon fixed frequency MF input with a binomial short-term plasticity model (*Rothman and Silver, 2014*; *Figure 4—figure supplement 2*). To probe the impact of tonic inhibitory conductance and $G_{NMDA}$ enhancement, we simulated four independent, Poisson-distributed MF inputs (*Figure 5F*) covering the wide range of MF frequencies observed in vivo (*Arenz et al., 2008*; *van Kan et al., 1993*). The model with GluA4-KO $G_{AMPA}$ and $G_{NMDA}$ and reduced tonic conductance displayed lower firing frequencies over the entire range of MF inputs (*Figure 5G*; *Figure 5—figure supplement 2*). Compared with WT, the KO model had an ~44% reduced maximum firing frequency (186.1 Hz vs. 336.4 Hz) and an ~17 Hz increased offset, similar to the experimental findings (*Figure 4C*). As in the current-clamp recordings (*Figure 4F*), spikes occurred with a longer delay in the KO GC model across all simulated MF frequencies (*Figure 5—figure supplement 2*).

To understand the individual contributions of reduced tonic inhibition and enhanced $G_{NMDA}$ to spiking in KO GCs, we either added an inhibitory conductance of 0.16 nS (calculated from *Figure 2F*) or used the WT $G_{NMDA}$ in the KO model. In both scenarios, firing frequencies were

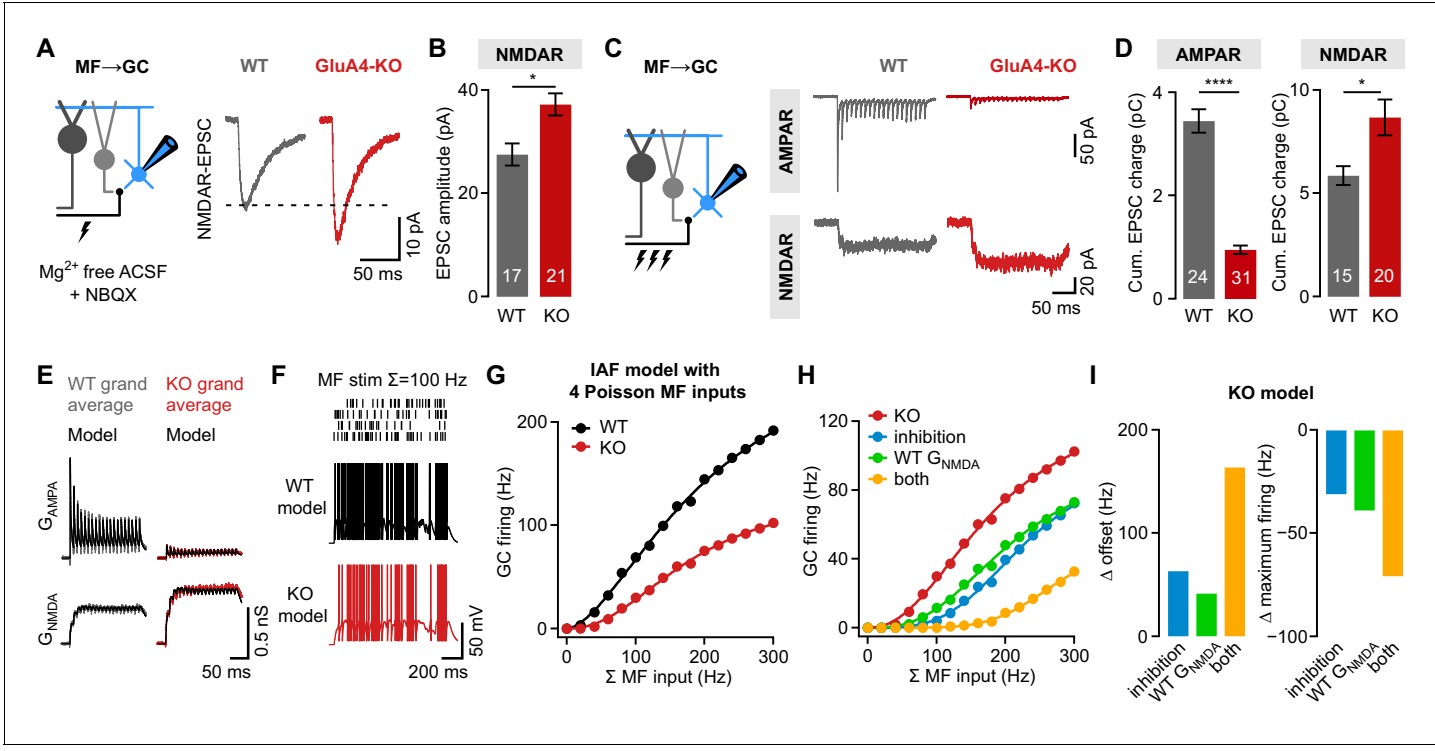

**Figure 5.** NMDA receptors (NMDARs) and reduced inhibition support spiking in GluA4-knockout (GluA4-KO) granule cells (GCs). (**A**) Left: Recordings of isolated NMDAR-EPSCs (excitatory postsynaptic currents) at mossy fiber (MF)→GC connections. Right: Example NMDAR-mediated MF→GC EPSCs. (**B**) Average EPSC amplitude for wild-type (WT) and GluA4-KO GCs (p = 0.0026; d = 1.04). (**C**) Left: Schematic of MF→GC train stimulation. Top: Example AMPA receptor (AMPAR)-EPSC 100 Hz train recordings (20 stimuli) for WT and KO. Bottom: Example NMDAR-mediated 100 Hz trains. (**D**) Average cumulative charge for AMPAR-mediated (left; p < 0.0001; d = −3.05) and NMDAR-mediated EPSCs (right; p = 0.0063; d = 0.92) for both genotypes. (**E**) Grand average of AMPAR and NMDAR conductance trains (20 stimuli, 100 Hz) recorded from WT (left) and GluA4-KO GCs (right). Data are overlaid with synaptic conductance models for WT and GluA4-KO. (**F**) Top: Four independent Poisson-distributed MF spike trains with a sum of 100 Hz (top) and simulation results for WT (black) and KO (red). (**G**) Average WT and KO model firing frequency vs. MF stimulation frequency. Lines are fits with a Hill equation. (**H**) Results for KO model compared with simulations of KO GCs with tonic inhibition (blue), with WT $G_{NMDA}$ (green) or both (yellow). (**I**) Reduced inhibition and enhanced $G_{NMDA}$ reduce the offset (left) and increase the maximum firing frequency (right) of the KO model GC output. Data are means ± SEM.

The online version of this article includes the following source data and figure supplement(s) for figure 5:

**Source data 1.** Numerical data plotted in *Figure 5*.

**Figure supplement 1.** Increased number of presynaptic release sites in GluA4-knockout (GluA4-KO) mice.

**Figure supplement 2.** Contribution of attenuated inhibition and enhanced $G_{NMDA}$ to granule cell (GC) spike output.

reduced in relation to the KO model, with each mechanism contributing ~20% to the maximum spiking frequency of the KO model (*Figure 5H–I*, *Figure 5—figure supplement 2*). Combining inhibition and WT $G_{NMDA}$ reduced the firing rate by ~40% (*Figure 5H–I*), implying a synergistic, linear interaction. We next asked how the experimentally observed changes in GluA4-KO GCs influence synaptic information transfer. Analysis of presynaptic and postsynaptic spike trains revealed that the strongly reduced $G_{AMPA}$ of the KO model compromises the synchrony of MF and GC spikes (approximately twofold reduction in spike synchronization *Mulansky et al., 2015*; *Figure 5—figure supplement 2*), and that the reduced tonic inhibition and $G_{NMDA}$ enhancement in GluA4-KO facilitate MF→GC synaptic information transfer at low input frequencies, albeit at a reduced level (*Figure 5G–H*, *Figure 5—figure supplement 2*). Thus, although MF→GC AMPAR-EPSCs are strongly reduced, synaptic and non-synaptic mechanisms cooperatively counteract the impaired spiking output of GluA4-KO GCs. However, both mechanisms are insufficient to compensate for the pronounced impairment of synaptic excitation caused by the deletion of GluA4.

## Impaired pattern separation in a feedforward model of the cerebellar input layer

GluA4 is essential for effective excitation of GCs (*Figures 1* and *5*), which form the input stage of the cerebellar cortex. GCs receive sparse input from only on average four MFs, and vastly outnumber afferent MFs, with an expansion ratio of ~200:1 (*Herculano-Houzel, 2009*). The abundance of GCs and their sparse connectivity make the GC layer ideally suited for sparsening and expanding of MF inputs, leading to a higher-dimensional representation of information (*Albus, 1971*; *Lanore et al., 2021*; *Marr, 1969*). This transformation of neural coding is thought to enable effective pattern separation and facilitate associative learning in downstream circuits (*Cayco-Gajic and Silver, 2019*). Because effective excitation of GCs ensures a high encoding capacity and lossless information transfer at the GC layer (*Billings et al., 2014*), deletion of GluA4 may compromise pattern separation and learning. To address how reduced synaptic input and cellular adaptations in GluA4-KO GCs affect information processing in the GC layer, we employed computational modeling of a feedforward MF→GC network with constrained connectivity and biophysical properties (*Figure 6A–B*; *Cayco-Gajic et al., 2017*). We systematically varied the fraction of active MFs and their spatial correlation, and analyzed how the GC network transforms MF activity patterns (*Cayco-Gajic et al., 2017*). To assess the impact of reduced synaptic excitation, we scaled model synaptic conductances and tonic inhibition according to our GluA4-KO data (see Materials and methods) (https://github.com/delvendahl/GluA4_cerebellum_eLife copy archived at *Delvendahl, 2021* swh:1:rev:1d1c19853e18f8fbb3307eeecc64096e53d74820). Both control and KO models showed a similar increase in spatial sparseness of GC activity patterns compared with MF activity patterns (*Figure 6C*), indicating that deletion of GluA4 did not affect population sparsening. In addition to sparsening, an important characteristic of the GC layer is an expansion of coding space, which promotes pattern separation by increasing the distance between activity patterns. To investigate expansion recoding, we analyzed the size of the distribution of activity patterns by calculating the total variance of GC population activity, normalized to the total variance of the MF population activity (*Cayco-Gajic et al., 2017*). The normalized total variance—reflecting the expansion of coding space—was strongly reduced in the GluA4-KO model (*Figure 6D*), suggesting that the increased spiking threshold of GluA4-KO GCs leads to a strong reduction of coding space at the cerebellar input layer.

To investigate the consequences of alterations in sparsening and expansion recoding for learning performance, we trained a perceptron to classify either MF or GC activity patterns into 10 random classes (*Figure 6E*), and analyzed learning speed using GC patterns normalized to MF patterns (*Cayco-Gajic et al., 2017*; *Figure 6F*). Learning speed was reduced over the full range of model parameters when using the GluA4-KO synaptic conductances (*Figure 6G*). Notably, GC activity accelerated learning only at high correlation levels and active fraction of MFs (white borders in *Figure 6G*), in contrast to the previously published model (*Cayco-Gajic et al., 2017*). This result is consistent with our current-clamp experiments, where higher MF stimulation frequencies elicited GC spikes in GluA4-KO mice (*Figure 4C* and *Figure 4—figure supplement 1*). Moreover, the total variance (*Figure 6D*), but not population sparseness, closely predicted learning speed, suggesting that the strong decrease of MF→GC transmission impairs learning speed by reducing the coding space. A decrease in the ratio of MF input strength to GC threshold might also reduce correlations in GC

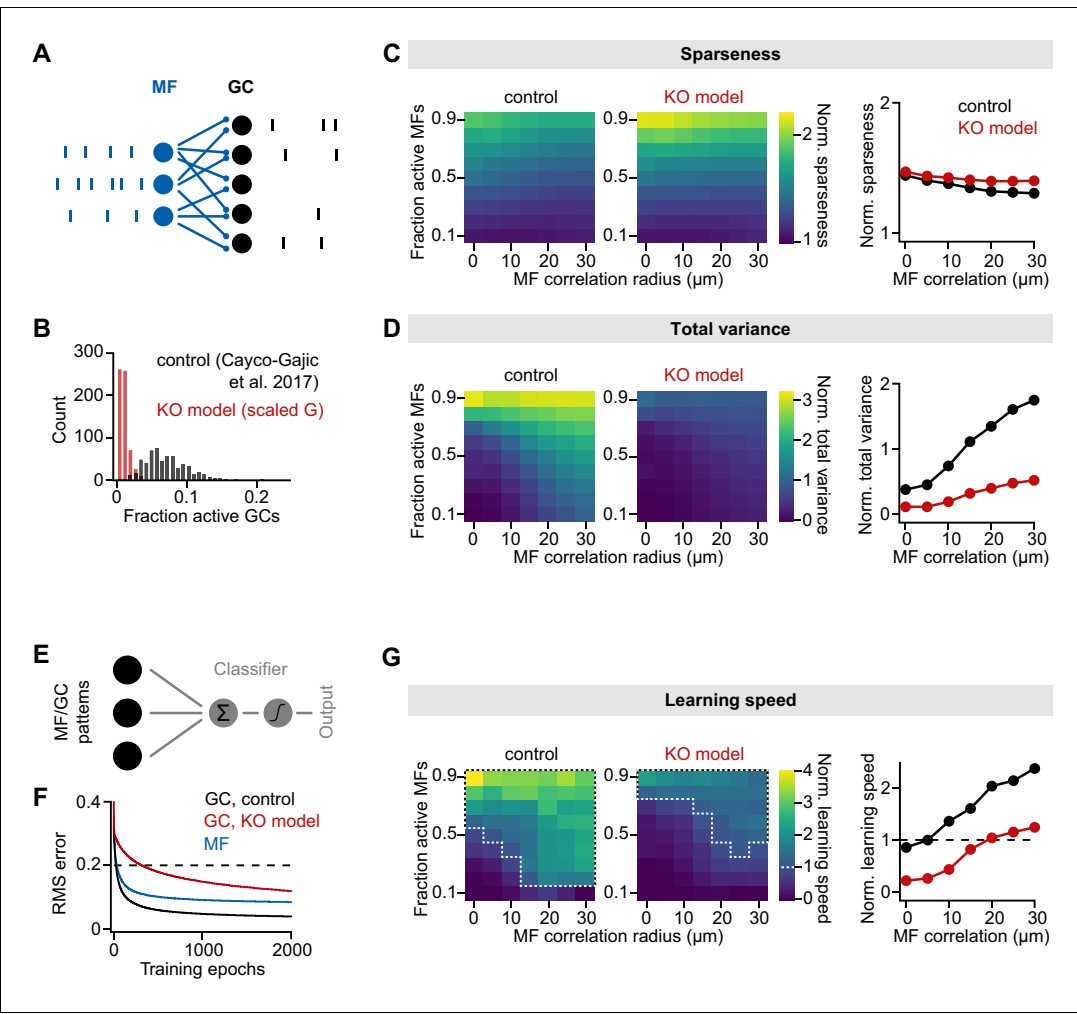

**Figure 6.** Impaired pattern separation in a feedforward model of the cerebellar input layer. (**A**) Schematic of the feedforward network model. The model comprises 187 mossy fibers (MFs) that are connected to 487 granule cells (GCs) with four synapses per GC. The network is presented with different MF activity patterns, which produce GC spike patterns. (**B**) Histogram of active GCs. Scaling synaptic (AMPA receptor [AMPAR] and NMDA receptor [NMDAR]) conductance and tonic inhibitory conductance according to the GluA4-knockout (GluA4-KO) data ('KO model', red) leads to strong reduction of active GCs compared to the model from *Cayco-Gajic et al., 2017* ('control', black). (**C**) Left: Normalized population sparseness (normalized to MF sparseness) plotted for different MF correlation radii and fraction of active MFs of both models. Right: Median normalized sparseness vs. MF correlation radius. Median is calculated across fractions of active MFs. (**D**) Same as (**C**), but for total variance. (**E**) Schematic of learning implementation. MF or GC activity patterns are used to train a perceptron decoder to classify these patterns into 10 random classes. (**F**) Root-mean-square (RMS) error of the perceptron classification for MF input patterns (blue), GC input patterns (black) or GC patterns with scaled conductance (red). Dashed line indicates cutoff for learning speed quantification. (**G**) Left: Normalized learning speed (normalized to learning using MF patterns) plotted for different MF correlation radii and fraction of active MFs of both models. White dashed borders indicate areas of faster learning with GC activity patterns than with MF patterns. Right: Median normalized learning speed vs. MF correlation radius. Dashed line indicates faster GC learning.

The online version of this article includes the following figure supplement(s) for figure 6:

**Figure supplement 1.** Population correlations in the mossy fiber (MF)-granule cell (GC) feedforward model.

output patterns (*Marr, 1969*). Indeed, population correlation was slightly reduced in the GluA4-KO model, but the stronger decorrelation had little effect on learning speed (*Figure 6—figure supplement 1*). Nevertheless, it is worth noting that a reduction in the fraction of active GCs per se might contribute to reduced learning speed in the GluA4-KO model. Together, our data indicate that

GluA4 facilitates expansion coding in a feedforward MF-GC network model without major effects on population sparseness. These effects on GC population coding may improve pattern separation and thus speed learning.

## Normal locomotor coordination of GluA4-KO mice during overground walking

Our results so far indicate that deletion of GluA4 drastically impairs MF→GC transmission and expansion coding, while the effects on GC spiking are less pronounced and population sparseness remains intact. We wondered how these alterations in information processing at the cerebellar input layer would impact cerebellum-dependent behavior. The cerebellum is critical for motor coordination and several forms of associative learning. We first analyzed the locomotor behavior of GluA4-KO mice. Cerebellar dysfunction often leads to gait ataxia during walking, characterized by altered 3D limb trajectories and deficits in interlimb and whole-body coordination in both humans and mice (*Machado et al., 2020a*; *Machado et al., 2015*; *Morton and Bastian, 2007*). Surprisingly, we observed that GluA4-KO mice walked similarly to size-matched WT littermates without obvious gait ataxia (*Video 1*). Quantitative analysis of locomotor coordination with the LocoMouse system (*Machado et al., 2020a*; *Machado et al., 2015*; *Figure 7A–C*) revealed that their locomotor behavior was largely intact. GluA4-KO mice tended to walk more slowly than controls, but across walking speeds, stride lengths were comparable in mice of both genotypes (*Figure 7D–E*, *Figure 7—figure supplement 1*). Paw trajectories measured by continuous forward velocity were also intact (*Figure 7F–G*, *Figure 7—figure supplement 1*). Vertical paw motion was also largely normal (*Figure 7H–I*), although there was a small degree of hyperflexion across walking speeds (*Figure 7— figure supplement 1*). We also explored interlimb coordination, which is often disrupted by cerebellar damage (*Machado et al., 2020a*; *Machado et al., 2015*; *Morton and Bastian, 2007*), during normal walking. We did not observe any differences between WT and GluA4-KO mice, with both genotypes displaying a symmetrical trot pattern across walking speeds (*Figure 7J–K*). Exhaustive analysis of the movements of individual limbs, interlimb and body coordination, and comparison with other mouse models of cerebellar ataxia confirmed the grossly normal locomotor phenotype of GluA4-KO mice (*Figure 7—figure supplements 1* and *2*). Moreover, GluA4-HET mice showed normal overall locomotion performance (*Figure 7—figure supplement 3*). Thus, locomotor coordination was largely spared in GluA4-KO mice despite the pronounced impairment of synaptic excitation at the cerebellar input layer.

## GluA4 is required for cerebellum-dependent associative memory formation

Influential theories of cerebellar function have proposed that pattern separation at the input layer facilitates associative learning within the cerebellar cortex (*Albus, 1971*; *Cayco-Gajic and Silver, 2019*; *Marr, 1969*). While the strong reduction of GC synaptic excitation did not impair locomotor coordination of GluA4-KO mice, it might specifically affect cerebellum-dependent associative learning. We therefore examined delay eyeblink conditioning in GluA4-KO mice, a cerebellum-dependent form of associative learning that involves MF inputs conveying the conditioned stimulus (CS) (*Albergaria et al., 2018*; *De Zeeuw and Yeo, 2005*; *Mauk, 1997*). A conditioning light pulse was repeatedly paired with an unconditioned air-puff stimulus (US) to the eye as described previously (*Albergaria et al., 2018*; *Figure 8A*). The US elicited a similar response in mice of both genotypes (p = 0.36; *Figure 8B*). Over training sessions, WT animals acquired a conditioned eyelid closure that preceded the US (*Figure 8C*). WT mice displayed a continuous increase in the percentage of trials with a conditioned response (CR; *Figure 8D*),

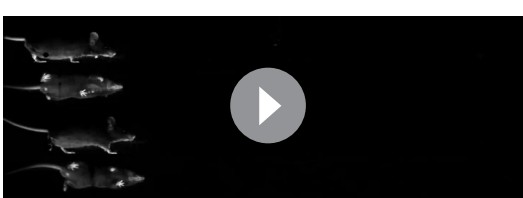

**Video 1.** Locomotion analysis of wild-type (WT) and GluA4-knockout (GluA4-KO) mice. High-speed (400 fps) video of a mouse crossing the LocoMouse corridor, displayed at 30 fps. Side and bottom (via mirror reflection) views of the mouse are captured in a single camera. Top: Raw video of a WT mouse. Bottom: Raw video of a GluA4-knockout (GluA4-KO) mouse.
https://elifesciences.org/articles/65152#video1

demonstrating associative memory formation. Strikingly, GluA4-KO mice failed to develop CRs over the 10 training sessions (*Figure 8C and D*, top), indicating a lack of associative learning in the eyeblink conditioning task. GluA4-HET animals, on the other hand, were able to acquire associative memories similarly to WT (*Figure 8—figure supplement 1*). Analysis of CS-only trials revealed that, in contrast to WT animals, which had developed well-timed CRs by the end of training, GluA4-KO mice failed to show any learned eyelid closures in response to the CS (*Figure 8E–F*, top). To rule out the possibility that impaired vision was contributing to this effect (*Gründer et al., 2000*; *Qin and Pourcho, 1999*), we repeated the eyeblink conditioning experiments in a new set of mice using whisker stimulation as a CS. Again, CRs were absent in GluA4-KO mice (*Figure 8C–F*, bottom), demonstrating that GluA4 is required for eyeblink conditioning independent of CS modality. Thus, GluA4 appears to be dispensable for the control of coordinated locomotion (*Figure 7*), but is required for normal cerebellum-dependent associative learning.

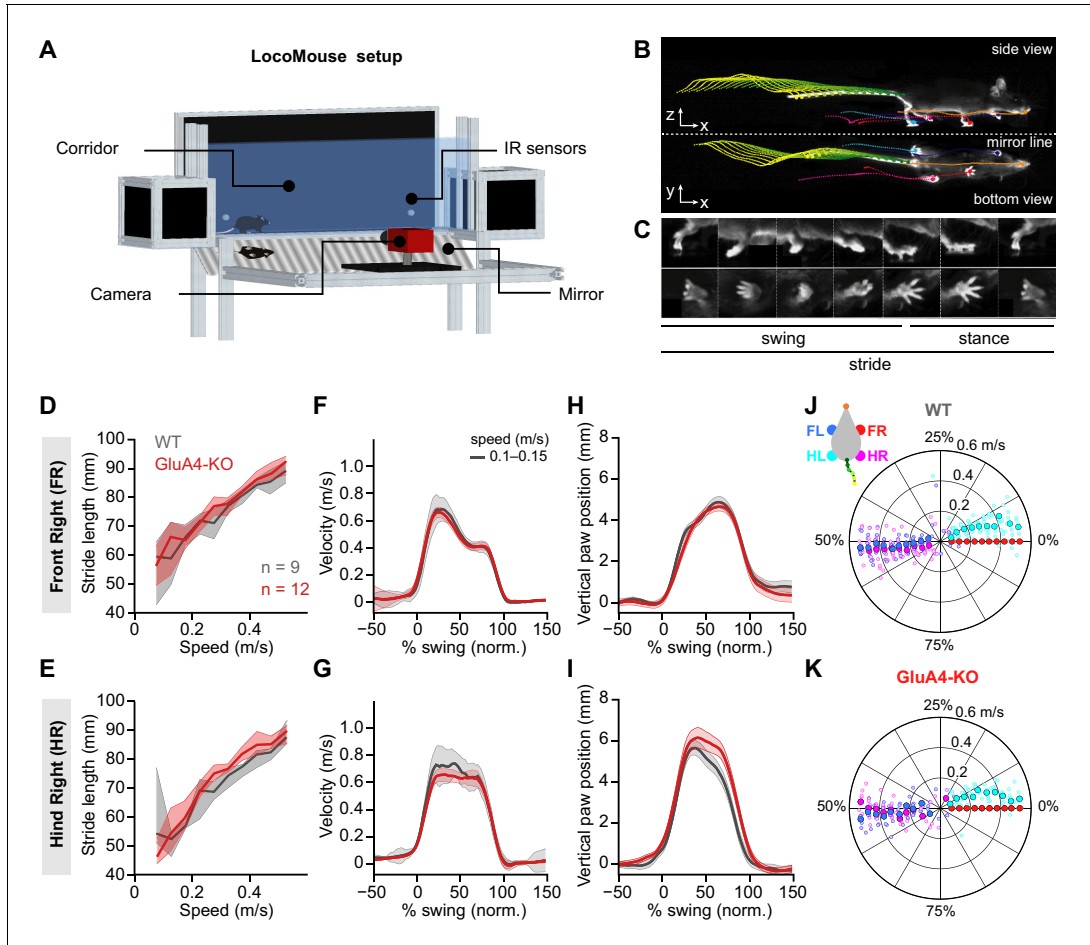

**Figure 7.** Normal locomotor coordination of GluA4-knockout (GluA4-KO) mice during overground walking. (**A**) LocoMouse apparatus. (**B**) Continuous tracks (in x, y, z) for nose, paws, and tail segments obtained from LocoMouse tracking (*Machado et al., 2015*) are plotted on top of the frame. (**C**) Individual strides were divided into swing and stance phases for further analysis. (**D**)–(**E**) Stride length of front right (FR) paw (**D**) and hind right (HR) paw (**E**) vs. walking speed for GluA4-KO mice (red) and wild-type (WT) animals (gray). For each parameter, the thin lines with shadows represent median values ± 25th, 75th percentiles. (**F**)–(**G**) Average instantaneous forward (x) velocity of FR paw (**F**) and HR paw (**G**) during swing phase. (**H**)–(**I**) Average vertical (z) position of FR paw (**H**) and HR paw (**I**) relative to ground during swing. The shaded area indicates SEM across mice. (**J**)–(**K**) Polar plots indicating the phase of the step cycle in which each limb enters stance, aligned to stance onset of FR paw (red circle). Distance from the origin represents walking speed. (**J**) WT mice; (**K**) GluA4-KO mice. Circles show average values for each animal.

The online version of this article includes the following figure supplement(s) for figure 7:

**Figure supplement 1.** Distribution of z-scored values for all gait parameters.

**Figure supplement 2.** Linear discriminant analysis separates ataxic mutants from GluA4-knockout (GluA4-KO) and control animals.

**Figure supplement 3.** Normal locomotion in heterozygous GluA4 (GluA4-HET) mice.

## Discussion

In the present study, we elucidated the functional significance of GluA4 in the cerebellum. Deletion of this AMPAR subunit strongly impaired excitatory transmission at the cerebellar input layer, expansion coding, and associative learning in adult mice. The drastic reduction in MF→GC transmission indicates a crucial role of GluA4 for GC excitation. In contrast, GC spike output was less reduced, due to compensatory enhancements in input-output relationship and NMDAR-mediated transmission. The synaptic and cellular alterations in GluA4-KO GCs resulted in compromised expansion coding of a feedforward circuit model without affecting population sparseness, suggesting that GluA4 supports pattern separation of cerebellar GCs. On a behavioral level, even though locomotor coordination was largely intact in GluA4-KO mice, they were unable to form associative memories during eyeblink conditioning. Together, our results demonstrate that GluA4 is required for reliable synaptic excitation at the cerebellar input layer and normal cerebellum-dependent associative learning.

### GluA4 mediates rapid synaptic excitation at the cerebellar input layer

GluA4 expression generally decreases during development (*Zhao et al., 2019*; *Zhu et al., 2000*), and previous studies on GluA4 were mainly performed during development or in young animals (*Fuchs et al., 2007*; *Huupponen et al., 2016*; *Yang et al., 2011*; *Zhu et al., 2000*; *Hadzic et al., 2017*, but see *Seol and Kuner, 2015*). The cerebellar cortex, however, retains high levels of GluA4 in the adult (*Keinänen et al., 1990*; *Monyer et al., 1991*; *Schwenk et al., 2014*), primarily due to the numerous GCs expressing this AMPAR subunit. Here, we show in adult mice that GluA4-containing AMPARs mediate the majority of excitatory input onto GCs and account for the fast kinetics of MF→GC EPSCs. These results provide additional evidence that the rapid kinetics of GluA4 is crucial for the precise timing of postsynaptic spikes (*Seol and Kuner, 2015*; *Yang et al., 2011*). Overall, synapses with high GluA4 expression are capable of high-frequency transmission and exhibit strong short-term depression. GluA4 may thus be an integral part of the molecular framework enabling high-frequency information transfer in the mammalian brain.

### Compensations in GluA4-KO mice

The strongly diminished MF→GC EPSCs in GluA4-KO mice indicate that other AMPAR subunits cannot compensate for the loss of GluA4 (*Yan et al., 2013*; *Yang et al., 2011*). We did, however, observe a reduction of tonic inhibition in GluA4-KO GCs, which may enhance information flow through the cerebellar cortex (*Hamann et al., 2002*). Our slice data of GC tonic inhibitory conductance are similar to observations in vivo (*Duguid et al., 2012*), where tonic inhibition controls GC excitability (*Chadderton et al., 2004*) and EPSC-spike coupling (*Duguid et al., 2012*). The reduced inhibition in GluA4-KO mice may thus enhance sensory information transfer at the cerebellar input layer. Intriguingly, we observed lower $\alpha6$ GABA$_A$R levels by Western blot analysis in the GluA4-KO cerebellum, consistent with previous results from *stargazer* and *waggler* mice (*Chen et al., 1999*; *Payne et al., 2007*). These findings suggest that reduced $\alpha6$ GABA$_A$R expression contributes to the modulation of GC tonic inhibition upon loss of AMPARs. Besides, increased NMDAR-EPSCs facilitated spiking of GluA4-KO GCs during high-frequency synaptic input. Compared with slice recordings, the contribution of NMDARs to GC synaptic excitation is likely to be larger in vivo (*Zhang et al., 2020*), due to the more depolarized membrane potential and increased spontaneous input in GCs. The larger NMDAR-EPSCs in GluA4-KO are most likely caused by enhanced glutamate release (*Chourbaji et al., 2008*; *Delvendahl et al., 2019*) that is driven by an increase in the number of presynaptic release sites, but we cannot exclude postsynaptic contributions. Together, attenuated inhibition and enhanced NMDAR-EPSCs contribute to synaptic information transfer by increasing the excitability of GluA4-KO GCs. Yet, these adaptations are insufficient for maintaining the fidelity of GC output at WT level. Additional compensations could occur at the level of morphology. While we did not observe obvious differences in cerebellar gross anatomy between WT and GluA4-KO mice, changes in, for instance, MF bouton ultrastructure or the number of MFs contacting a GC could affect the properties of GCs and MF→GC synapses in GluA4-KO mice.

### A low number of active GCs can sustain normal locomotion

Despite the large reduction of MF→GC transmission, locomotor coordination was not impaired. This may point toward other pathways contributing to sensorimotor coordination or could be due to

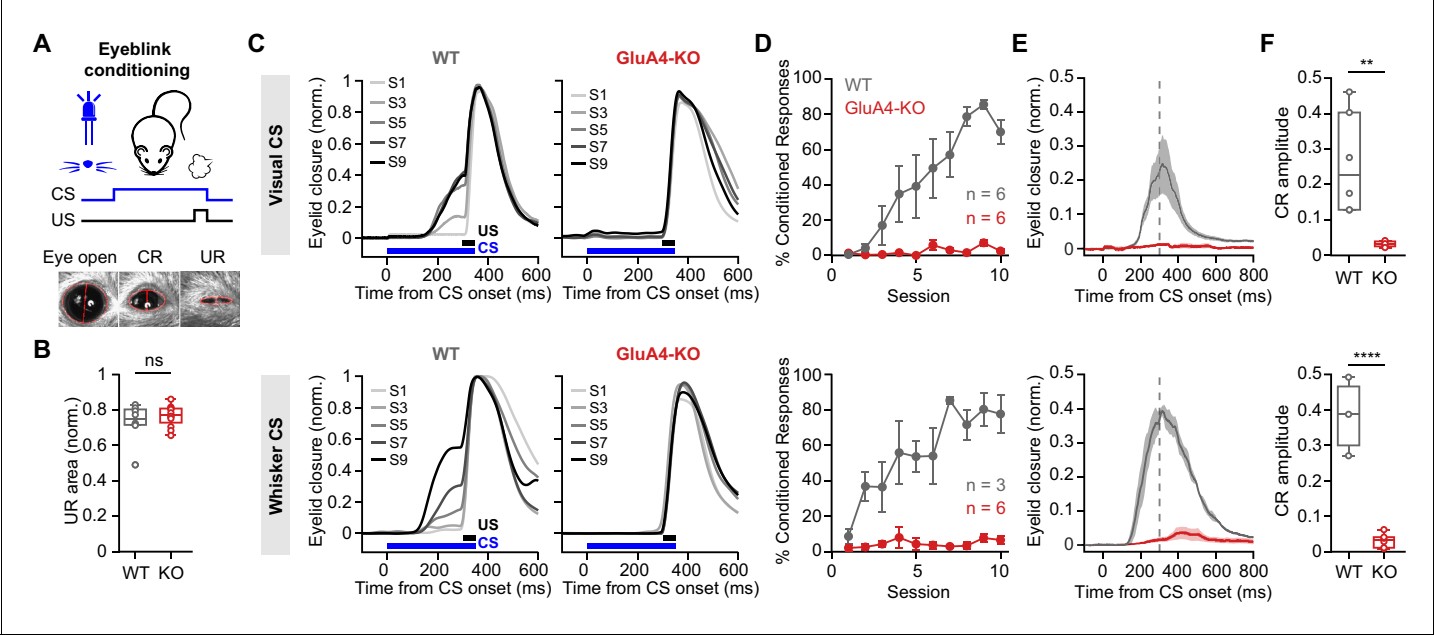

**Figure 8.** GluA4 is required for cerebellum-dependent associative memory formation. **(A)** Top: Schematic of the eyeblink conditioning paradigm, where a conditioned stimulus (CS) (a white LED or a whisker vibratory stimulus) is repeatedly paired and co-terminates with an eyeblink-eliciting unconditioned stimulus (US) (a puff of air delivered to the eye). Bottom: Example video frames (acquired at 900 fps under infrared light) illustrate automated extraction of eyelid movement amplitude. **(B)** Normalized mean unconditioned response area for wild type (WT) and GluA4-knockout (GluA4-KO) (WT: n = 12; KO: n = 9; p = 0.36). **(C)** Average eyelid closure across nine learning sessions (S1–S9) of representative WT and KO animals, using a visual CS (top) or a whisker CS (bottom). Each trace represents the average of 100 paired trials from a single session. **(D)** Average %CR (% conditioned response) learning curves of WT (gray) and GluA4-KO (red) mice trained with either a visual CS (top) or a whisker CS (bottom). Error bars indicate SEM. **(E)** Average eyelid traces of CS-only trials from the last training session of WT and GluA4-KO animals, trained with either a visual CS (top) or a whisker CS (bottom). Shadows indicate SEM. Vertical dashed line represents the time that the US would have been expected on CS+US trials. **(F)** Average CR amplitudes from the last training session of WT and GluA4-KO animals, trained with either a visual CS (top; p = 0.0029) or a whisker CS (bottom; p < 0.0001). Box indicates median and 25th–75th percentiles; whiskers extend to the most extreme data points.

The online version of this article includes the following figure supplement(s) for figure 8:

**Figure supplement 1.** Normal associative learning in heterozygous GluA4 (GluA4-HET) mice.

---

developmental compensation. It is intriguing to speculate that attenuated tonic inhibition and enhanced non-AMPAR transmission sustain sufficient levels of MF→GC transmission in GluA4-KO animals to spare locomotor activity. The raised GC threshold might also be overcome by activation of multiple MF inputs, which can increase GC spike rates in vivo (*Ishikawa et al., 2015*). MF activity during locomotion is generally dense (*Ishikawa et al., 2015*; *Knogler et al., 2017*; *Powell et al., 2015*) and may thus activate a sufficient number of GCs for locomotor coordination in GluA4-KO mice. By contrast, the MF representation of the CS used for eyeblink conditioning is most likely not widespread and dense enough to activate a critical fraction of GCs in GluA4-KO mice (*Giovannucci et al., 2017*; *Shimuta et al., 2020*). Our finding that in the KO model, GC activity patterns promote faster learning than MF inputs only for high levels of MF activity might thus explain the discrepancy between spared locomotion and absence of associative learning in GluA4-KO mice. These results are in line with previous studies, where impairments of MF→GC or PF→PC synapses did not cause locomotion defects but affected motor learning (*Andreescu et al., 2011*; *Galliano et al., 2013*; *Peter et al., 2020*; *Seja et al., 2012*). Indeed, a small number of active GCs may be sufficient for basic motor performance (*Galliano et al., 2013*; *Schweighofer et al., 2001*).

Our slice recordings were confined to cerebellar lobules III–VI and regional differences in GluA4 expression could in principle contribute to the preserved locomotion in GluA4-KO mice. However, GluA4 expression appears uniform across different lobules within the cerebellar cortex (*Yamasaki et al., 2011*; Allen Brain Atlas), and lobules III–VI are strongly involved in locomotion (*Giovannucci et al., 2017*; *Markwalter et al., 2019*; *Muzzu et al., 2018*; *Powell et al., 2015*). We

therefore consider it unlikely that regional differences play a major role in the differential effects on eyeblink conditioning vs. locomotion in GluA4-KO animals.

## The role of MF→GC synapses in associative learning

Several previous studies investigated PF→PC synaptic function and plasticity in cerebellum-dependent motor learning (*Galliano et al., 2013*; *Grasselli et al., 2020*; *Gutierrez-Castellanos et al., 2017*; *Peter et al., 2020*; *Wada et al., 2007*). Afferent sensory information to the cerebellar cortex is first processed at the upstream MF→GC synapse (*Billings et al., 2014*; *Cayco-Gajic et al., 2017*), which may support cerebellar learning. Indeed, disturbed AMPAR trafficking at MF→GC, as well as other cerebellar synapses, in ataxic *stargazer* mice compromises eyeblink conditioning (*Hashimoto et al., 1999*; *Jackson and Nicoll, 2011*). Our findings in GluA4-KO mice further support a role of MF→GC synapses in cerebellar associative learning. We note that by using a systemic KO model, a functional role of the GluA4 AMPAR subunit at other synapses within the circuits responsible for the behavioral tasks assessed in our study may contribute to the observed results. However, the expression of GluA4 is generally low in the mammalian CNS outside of the cerebellum (*Hadzic et al., 2017*; *Schwenk et al., 2014*; *Sjöstedt et al., 2020*; *Yamasaki et al., 2011*). We show here that this AMPAR subunit is crucial for synaptic excitation of cerebellar GCs, but not of PCs and GoCs. Besides, GluA4 is expressed in Bergmann glia, but deletion of AMPARs in these cells does not affect eyeblink conditioning (*Saab et al., 2012*). Although we cannot rule out a contribution from other GluA4-containing synapses elsewhere in the brain, associative memory formation during eyeblink conditioning takes place within the cerebellum (*Carey, 2011*; *De Zeeuw and Yeo, 2005*; *Freeman and Steinmetz, 2011*; *Heiney et al., 2014*; *Mauk, 1997*; *McCormick and Thompson, 1984*), where GluA4 is predominantly expressed in GCs. The finding that different sensory modalities (visual and somatosensory) lead to similar deficits in associative learning and that unconditioned responses to the air-puff stimulus are intact in GluA4-KO mice further argue for a major role of cerebellar GluA4 for the behavioral results. Thus, taking the limitations of a systemic KO model into account, our results indicate that GluA4-mediated MF→GC transmission is essential for cerebellar associative learning. At this synaptic connection, the properties of the GluA4 AMPAR subunit are likely to be important for transmission and processing of the CS, which is conveyed by MF inputs (*Albergaria et al., 2018*; *De Zeeuw and Yeo, 2005*; *Mauk, 1997*).

## GluA4 supports computations underlying cerebellar learning

The remaining level of GC excitation in the absence of GluA4 seems sufficient for basic motor performance but does not allow for associative learning. Despite the ~80% reduction in MF→GC EPSCs, spike output upon MF stimulation was only reduced by ~50% in GluA4-KO GCs, and the sparseness of GC population activity was not impaired. These findings indicate that locomotor coordination can be sustained by low levels of GC-population activity, and may rely on effective population sparsening. In contrast, associative memory formation may require more complex levels of GC activity and/or synaptic integration (*Albergaria et al., 2018*; *Carey, 2011*; *Raymond and Medina, 2018*). It is interesting to note that deletion of GluA4 not only caused a strong overall reduction in GC firing frequency, but also severely prolonged the first spike delay upon MF stimulation. GCs are subject to feedback and feedforward inhibition by GoCs, which narrows the time window for GC integration to ~5–10 ms (*D'Angelo and De Zeeuw, 2009*). The longer spike delay in GluA4-KO mice, together with the reduced MF→GC synaptic strength, raises the threshold of GC activation, which leads to a loss of information (*Billings et al., 2014*; *Cayco-Gajic et al., 2017*). Due to a longer delay, GC spikes upon onset of the CS might also arrive too late to trigger a CR. Furthermore, the alterations of MF→GC transmission in the absence of GluA4 could reduce the reliability and increase the temporal variability of PF-driven PC excitation in response to sensory stimulation, which may interfere with synaptic plasticity at PF→PC synapses (*Suvrathan and Raymond, 2018*). Our modeling results suggest that the higher GC threshold impaired expansion recoding of MF inputs onto the large population of GCs (*Albus, 1971*; *Marr, 1969*) by effectively reducing the coding space. By contrast, the population sparseness of GCs was largely unaffected. Reduced expansion coding is expected to impair pattern separation and learning in downstream circuits. On the other hand, the effective rise of the GC threshold due to weak MF input strength may by itself decrease overlap between GC outputs, which would be predicted to improve learning speed (*Marr, 1969*). Together, our findings are

consistent with the hypothesis that pattern separation at the cerebellar input layer promotes associative memory formation.

The results we present here provide further evidence for the concept of an AMPAR code (*Diering and Huganir, 2018*), in which different AMPAR subunits serve different roles for synaptic function and learning. Future studies exploring the possible involvement of GluA4 in synaptic plasticity and other forms of learning will be an important component of our efforts to understand the rules governing cell-type-specific expression and function of AMPAR subunits in the CNS.

# Materials and methods

## Key resources table

| Reagent type (species) or resource | Designation | Source or reference | Identifiers | Additional information |
|---|---|---|---|---|
| Strain, strain background (*Mus musculus*, ♀ and ♂) | Gria4$^{tm1.1Mony}$/Gria4$^{tm1.1Mony}$ ('GluA4-KO') | *Fuchs et al., 2007* | RRID:MGI:3804915 | |
| Antibody | Anti-GluA4 F-9 (mouse monoclonal) | Santa Cruz Biotechnology | Cat. # sc-271894 RRID:AB_10715096 | WB (1:100–1:200) |
| Antibody | Anti-GABA$_A$R-α6 (rabbit polyclonal) | Alomone | Cat. # AGA-004 RRID:AB_2039868 | WB (1:500) |
| Antibody | Anti-GAPDH GA1R (mouse monoclonal) | Thermo Fisher Scientific | Cat. # MA5-15738-1MG RRID:AB_2537652 | WB (1:2000) |
| Antibody | Anti-β-tubulin (mouse monoclonal) | Developmental Studies Hybridoma Bank | Cat. # E7 RRID:AB_2315513 | WB (1:1000) |
| Chemical compound, drug | Bicuculline methiodide | Sigma-Aldrich | Cat. # 14343 | |
| Chemical compound, drug | Strychnine | Sigma-Aldrich | Cat. # S8753 | |
| Chemical compound, drug | GYKI-53655 | Tocris | Cat. # 2555 | |
| Chemical compound, drug | NBQX | HelloBio | Cat. # HB0443 | |
| Chemical compound, drug | D-APV | Tocris | Cat. # 0106 | |
| Chemical compound, drug | TTX | Tocris | Cat. # 1069 | |
| Software, algorithm | Igor Pro | WaveMetrics | RRID:SCR_000325 | Version 6.37 |
| Software, algorithm | NeuroMatic | *Rothman and Silver, 2018* | RRID:SCR_004186 | Version 3.0 c |
| Software, algorithm | R | The R Foundation | RRID:SCR_001905 | Version 3.6.3 |
| Software, algorithm | *lme4* | *Bates et al., 2015* | RRID:SCR_001905 | Version 1.0–5 |
| Software, algorithm | *effsize* | https://github.com/mtorchiano/effsize/ *Torchiano et al., 2020* | | Version 0.8.1 |
| Software, algorithm | *sjstats* | https://strengejacke.github.io/sjstats/ *Lüdecke, 2018* | | Version 0.18.1 |
| Software, algorithm | Python | Python Software Foundation | RRID:SCR_008394 | Version 2.7 or 3.7 |
| Software, algorithm | ImageJ | NIH | RRID:SCR_003070 | Version 2.1.0 |
| Other | Borosilicate glass | Science Products | Cat. # GB150F-10P | Outer/inner diameter: 1.5/0.86 mm |

## Animals

Animals were treated in accordance with national and institutional guidelines. All experiments were approved by the Cantonal Veterinary Office of Zurich (authorization no. ZH206/2016 and ZH009/

2020) or by the Portuguese Direcção Geral de Veterinária (Ref. No. 0421/000/000/2015). GluA4-KO mice were kindly provided by H Monyer (*Fuchs et al., 2007*). GluA4-KO (*Gria4$^{-/-}$*), GluA4-HET (*Gria4$^{+/-}$*), and WT (*Gria4$^{+/+}$*) littermates were bred from heterozygous crosses. Genotyping of GluA4-KO mice was performed as described (*Delvendahl et al., 2019*). Experiments were performed in male and female mice, 1–5 months of age for slice recordings and 2–5 months of age for behavioral experiments. The animals were housed in groups of three to five in standard cages with food and water ad libitum; they were kept on a 12h-light/12h-dark cycle that was reversed for behavioral experiments. No explicit power analysis was used; sample sizes were chosen based on previous similar experiments. Individual recordings were treated as independent samples for slice electrophysiology and animals for behavioral experiments. No blinding or randomization was used.

## Electrophysiology

Mice were sacrificed by rapid decapitation either without prior anesthesia or after isoflurane anesthesia in later experiments according to national guidelines. The cerebellar vermis was removed quickly and mounted in a chamber filled with cooled extracellular solution; 300-µm-thick parasagittal slices were cut using a Leica VT1200S vibratome (Leica Microsystems, Germany), transferred to an incubation chamber at 35°C for 30 min and then stored at room temperature until experiments. The extracellular solution (artificial cerebrospinal fluid, ACSF) for slice cutting and storage contained (in mM): 125 NaCl, 25 NaHCO$_3$, 20 D-glucose, 2.5 KCl, 2 CaCl$_2$, 1.25 NaH$_2$PO$_4$, 1 MgCl$_2$, bubbled with 95% O$_2$, and 5% CO$_2$. For recordings from GoCs, the slicing solution consisted of (in mM): 230 sucrose, 25 D-glucose, 24 NaHCO$_3$, 4 MgCl$_2$, 2.5 KCl, 1.25 NaH$_2$PO$_4$, 0.5 CaCl$_2$, 0.02 D-APV.

Cerebellar slices were visualized using an upright microscope with a 60×, 1 NA water immersion objective, infrared optics, and differential interference contrast (Scientifica, UK). The recording chamber was continuously perfused with ACSF supplemented with 10 µM D-APV, 10 µM bicuculline, and 1 µM strychnine unless otherwise stated. Voltage- and current-clamp recordings were done using a HEKA EPC10 amplifier (HEKA Elektronik GmbH, Germany). Data were filtered at 10 kHz and digitized with 100–200 kHz; recordings of spontaneous postsynaptic currents were filtered at 2.7 kHz and digitized with 50 kHz. Experiments were performed at room temperature (21–25°C). Patch pipettes were pulled to open-tip resistances of 3–8 MΩ (when filled with intracellular solution) from borosilicate glass (Science Products, Germany) using a DMZ puller (Zeitz Instruments, Germany). The intracellular solution for EPSC and current-clamp recordings contained (in mM): 150 K-D-gluconate, 10 NaCl, 10 HEPES, 3 MgATP, 0.3 NaGTP, 0.05 ethyleneglycol-bis(2-aminoethylether)-*N,N,N',N'*-tetraacetic acid (EGTA), pH adjusted to 7.3 using KOH. Voltages were corrected for a liquid junction potential of +13 mV.

We recorded in lobules III–VI of the cerebellar vermis. GC recordings and measurements of AMPAR-mediated MF→GC EPSCs were performed essentially as described previously (*Delvendahl et al., 2019*; *Delvendahl et al., 2015*). NMDAR-mediated MF→GC EPSCs were recorded at a holding potential of –80 mV using a Mg-free extracellular solution supplemented with 10 µM NBQX. GC excitability was assessed in current-clamp mode by tonic current injections (duration, 200 ms). For each sweep, the current amplitude was incremented by 5 pA, until a maximum of 40 pA. Current injections were applied from the GC resting membrane potential, which was not different between KO and WT ($-97.1 \pm 0.6$ mV, n = 37 vs. $-97.8 \pm 0.6$ mV, n = 46; p = 0.46). Action potential firing was quantified over the full duration of current injection.

To record sIPSCs and tonic inhibitory currents in GCs, we used a CsCl-based intracellular solution containing (in mM): 135 CsCl, 20 TEA-Cl, 10 HEPES, 5 Na$_2$phosphocreatine, 4 MgATP, 0.3 NaGTP, 0.2 EGTA, pH adjusted to 7.3 using CsOH (liquid junction potential ~0 mV). Due to the high intracellular [Cl$^-$], IPSCs were recorded as inward currents at a holding potential of –80 mV; 20 µM GYKI-53655 were added to the extracellular solution to block AMPARs. To quantify tonic GABA$_A$R-mediated conductance, recordings were made in the presence of 20 µM GYKI-53655 and 1 µM TTX. Following a stable baseline period, 10 µM bicuculline were bath-applied. Tonic conductance was calculated from the difference in holding current before and after bicuculline application, and normalized to cell capacitance to account for cell-to-cell variability. GC whole-cell capacitance was not different between genotypes (*Supplementary file 3*).

To study GC action potential firing upon high-frequency MF stimulation, membrane voltage was maintained at $-70$ mV (*Rothman et al., 2009*) by current injection. Similar results were obtained at a membrane potential of $-80$ mV (*Figure 4—figure supplement 1*). Ten MF stimuli were applied at

frequencies of 100–300 Hz. GC spikes were detected using a threshold of −20 mV and spike frequency was calculated over the entire train duration.

GoCs were identified based upon their position in the GC layer, their large capacitance (34.8 ± 3.9 pF, n = 29) and their low frequency of action potential firing (*Dieudonné, 1995*; *Kanichay and Silver, 2008*). Spontaneous EPSCs were recorded at a holding potential of –80 mV. MF→GoC EPSCs were recorded upon stimulation of the white matter or lower GC layer and identified based on the short latency and rapid kinetics of EPSCs and the absence of short-term facilitation (*Kanichay and Silver, 2008*; *Figure 1—figure supplement 1*). PF→GoC EPSCs were recorded upon stimulation of the molecular layer and displayed a longer, variable delay, slower kinetics, and pronounced short-term facilitation (*Kanichay and Silver, 2008*; *Figure 1—figure supplement 1*). Recordings from PCs were made similarly; PF→PC EPSCs were recorded upon stimulation of the molecular layer with increasing stimulation voltages (3–18 V) and spontaneous EPSCs were recorded at a holding potential of –80 mV.

## Western blotting

Cerebellar tissue was rapidly dissected and frozen at −80°C. Samples were lysed and sonicated in RIPA buffer with protein inhibitors (Roche). Protein concentrations were estimated by BCA assay (Thermo Fisher Scientific). Protein extracts were then denatured with 2× Laemmi buffer and boiled at 95°C for 5 min. Samples with equal amount of protein were run on 10% or 12% SDS-polyarylamide gel, and then transferred onto PVDF membrane (Thermo Fisher Scientific). Membranes were cut horizontally at appropriate molecular weights and blocked in 5% milk (w/v) in 1× PBS or TBST (0.1% Tween) for 1 hr on a shaker. Blots were then incubated with primary antibodies (anti-GluA4, anti-α6 GABA$_A$R, anti-GAPDH, or anti-β-tubulin) overnight at 4°C. To detect transferred proteins, we used horseradish peroxide-conjugated goat anti-mouse or anti-rabbit antibodies (1:1000–5000, Jackson Immuno Research). Blots were developed with a chemiluminescence detection kit (Thermo Fisher Scientific) and images were acquired with Fusion SL (Vilber). Band intensities were quantified using ImageJ software and normalized to GAPDH or β-tubulin signal.

## MF→GC modeling

A simple GC adaptive exponential integrate-and-fire model (*Brette and Gerstner, 2005*) was implemented and run using Neuromatic (*Rothman and Silver, 2018*) in Igor Pro (WaveMetrics, Tigard, OR) with a fixed time step of 50 μs. Model parameters were constrained using our experimental data or taken from the literature. To model MF→GC synapses, population averages of 100 Hz AMPAR and NMDAR trains were converted into synaptic conductance with direct and spillover components (*Rothman et al., 2009*). We optimized the model $G_{AMPA}$ and $G_{NMDA}$ parameters to fit the synaptic conductance during 100 Hz trains. Short-term plasticity was implemented using an R*P model (*Rothman and Silver, 2018*) and parameters were fine-tuned using 100 and 300 Hz population average data for each genotype. Model parameters are given in *Supplementary file 2*. Four independent Poisson spike trains were applied to the GC model, with a sum of 10–1000 Hz. Qualitatively, similar results were obtained with lower fractions of active MFs. Duration of the spike trains was 1 s and each MF input had a refractory time of 0.6 ms (*Ritzau-Jost et al., 2014*). Simulations were performed for 37°C and resting membrane potential was set at −80 mV. The average model spiking frequency was calculated from simulating 10 independent runs for each frequency. Plots of GC firing vs. MF stimulation frequency were fit with a Hill equation:

$$F(x) = \frac{F_{max}}{1 + \left(\frac{x_{1/2}}{x}\right)^n}$$

where $F_{max}$ is the maximum firing rate, $x_{1/2}$ the MF stimulation frequency at which F reaches half maximum, and n the exponent factor. The offset was taken from $x_{1/2}$ of the fits; the gain was calculated from the slope of the fits between 2% and 60% of the maximum (*Rothman et al., 2009*).

## Feedforward GC layer network modeling

We adapted the model described in *Cayco-Gajic et al., 2017* to study the effect of GluA4-KO on the function of the cerebellar input layer. In the model, 187 MFs are connected to 487 GCs in a sphere of 80 μm diameter; neuron numbers and connectivity are based on anatomical data. The

feedforward model was presented with n = 640 different MF input patterns (50 Hz firing, sampled from a Poisson distribution) and GC activity (i.e. spike counts in a 30 ms time window) was extracted. A perceptron algorithm was subsequently used to classify MF and GC activity patterns into 10 random classes (*Cayco-Gajic et al., 2017*). Input patterns were connected to the perceptron as described in *Cayco-Gajic et al., 2017*. We compared the published model parameters (*Cayco-Gajic et al., 2017*) against a model with scaled $G_{AMPA}$ and $G_{NMDA}$ corresponding to the observed relative changes in GluA4-KO GCs (scaling factors 0.2 and 1.2, respectively, *Figures 1C* and *5B*). In addition, the input conductance of the GluA4-KO model was reduced by 0.16 nS (according to *Figure 2G*). Learning speed, population sparseness, and total variance were calculated and analyzed as described in *Cayco-Gajic et al., 2017*. In brief, learning speed was analyzed as inverse of the number of training epochs needed to reach a root-mean-square error of 0.2. Population sparseness was calculated as:

$$\frac{N - \frac{\left(\sum_i x_i\right)^2}{\sum_i x_i^2}}{N - 1}$$

where N is the number of GCs and $x_i$ is the ith GC's spike count. Sparseness was averaged over all n activity patterns. Total variance was calculated as the sum of all variances:

$$\sum_i var(\boldsymbol{v_i})$$

where $\boldsymbol{v_i}$ is a vector of length n consisting of the ith GC's spike counts for all n activity patterns. Population correlation was assessed using the covariance as described in *Cayco-Gajic et al., 2017*.

## Locomotion analysis

We used the LocoMouse system to quantify overground locomotion (*Machado et al., 2015*). Animals run back and forward between two dark boxes, in a glass corridor (66.5 cm long and 4.5 cm wide with a mirror placed at 45° under). Individual trials consisted of single crossings of the corridor. A single high-speed camera (AVT Bonito, 1440 × 250 pixels, 400 fps) recorded both bottom and side views of walking mice during the trial. Animals were acclimated to the overground setup for several sessions before data collection; no food or water restriction or reward was used.

Homozygous GluA4-KOs ranged in size from 16 to 37 g. Size-matched littermate HET (18–38 g) and WT animals (16–38 g) were selected as controls (*Machado et al., 2015*). Ten to twenty-five corridor trials were collected in each session for 5 consecutive days. An average of 1207 ± 203 strides were collected per homozygous GluA4 mouse (310 ± 51 strides per animal per paw), 856 ± 342 strides per heterozygous GluA4 mouse (219 ± 94 strides per animal per paw), and 840 ± 278 strides per WT mouse (218 ± 70 strides per animal per paw) were collected.

## Delay eyeblink conditioning

Animals were anesthetized with isoflurane (4% induction and 0.5–1% for maintenance), placed in a stereotaxic frame (David Kopf Instruments, Tujunga, CA), and a head plate was glued to the skull with dental cement (Super-Bond C&B). After the surgery, mice were monitored and allowed at least 2 days of recovery.

All eyeblink conditioning experiments were run on a motorized treadmill, as described in previous work (*Albergaria et al., 2018*). Briefly, head-fixed mice were habituated to the motorized behavioral setup for at least 2 days prior to training. The speed of the treadmill was set to 0.12 m/s using a DC motor with an encoder (Maxon). After habituation, each training session consisted of 100 trials, separated by a randomized inter-trial interval of 10–15 s. In each trial, CS and US onsets were separated by a fixed interval of 300 ms and both stimuli co-terminated.

For all training experiments, the US was an air-puff (30–50 psi, 50 ms) controlled by a Picospritzer (Parker) and was adjusted for each session of each mouse so that the US elicited a full eyeblink. The CS had a 350 ms duration and was either (i) a white light LED positioned ~3 cm directly in front of the mouse or (ii) a piezoelectric device placed ~0.5 cm away from the left vibrissal pad.

Eyelid movements of the right eye were recorded using a high-speed monochromatic camera (Genie HM640, Dalsa) to monitor a 172 × 160 pixel region, at 900 fps. Custom-written software

using LabVIEW, together with an NI PCIE-8235 frame grabber and an NI-DAQmx board (National Instruments), was used to trigger and control all the hardware in a synchronized manner. Mean UR area was analyzed for training session 1. We calculated area under the eyelid movement curve between time of US and US+300 ms, and normalized data to the maximum area of that session for each mouse.

## Data analysis

Electrophysiological data were analyzed using custom-written routines or Neuromatic (*Rothman and Silver, 2018*) in Igor Pro (WaveMetrics, Tigard, OR). EPSC amplitudes were quantified as difference between peak and baseline. Decay kinetics were analyzed by fitting a bi-exponential function to the EPSC decay time course. To detect spontaneous or miniature postsynaptic currents, we used a template-matching routine (*Rothman and Silver, 2018*). Detected events were analyzed as described above. Only cells with >10 detected spontaneous events were included for amplitude quantification. GC excitability was assessed using gain and rheobase of the input-output relationship. Rheobase was defined as minimum current injection amplitude eliciting at least a single action potential. Gain was calculated from linear fits to the input-output relationship. Average gain in WT was $4.3 \pm 0.4$ Hz (control, n = 36 GCs), in line with previous studies (*Gall et al., 2003*; *Rizwan et al., 2016*; *Rudolph et al., 2020*; *Soda et al., 2019*; *Straub et al., 2020*). For display purposes, traces in figures were smoothed with a Savitzky-Golay second-order filter with a 25-point window size.

Simulation results were analyzed using Igor Pro and Python. To calculate the van Rossum distance, pre- and postsynaptic spike trains were convolved using a 10 ms exponential kernel. The area of the squared difference was then multiplied by the inverse of the kernel time constant. Spike synchronization was calculated using PySpike (*Mulansky and Kreuz, 2016*).

For eyeblink conditioning, video from each trial was analyzed offline with custom-written software using MATLAB (MathWorks). The distance between eyelids was calculated frame by frame by thresholding the grayscale image of the eye and extracting the count of pixels that constitute the minor axis of the elliptical shape that delineates the eye. Eyelid traces were normalized for each session, ranging from 1 (full blink) to 0 (eye fully open). Trials were classified as CRs if the eyelid closure reached at least 0.1 (in normalized pixel values) and occurred between 100 ms after the time of CS onset and the onset of US.

Analysis of locomotion data was performed in MATLAB 2012b and 2015a. Paw, nose, and tail tracks (x,y,z) were obtained from the LocoMouse tracker (*Machado et al., 2015*) (https://github.com/careylab/LocoMouse [*Machado et al., 2020b*]). All tracks were divided into stride cycles and the data was sorted into speed bins (0.05 m/s binwidth). Gait parameters (individual limb movements and interlimb coordination) were calculated as follows:

### Individual limb

Walking speed: x displacement of the body center during that stride divided by the stride duration.
Stride length: x displacement from touchdown to touchdown of single limb.
Stride duration: time between two consecutive stance onsets.
Cadence: inverse of stride duration.
Swing velocity: x displacement of single limb during swing phase divided by swing duration.
Stance duration: time in milliseconds that foot is on the ground during stride.
Duty factor: stance duration divided by stride duration.
Trajectories: (x,y,z) trajectories were aligned to swing onset and resampled to 100 equidistant points using linear interpolation. Interpolated trajectories were then binned by speed and the average trajectory was computed for each individual animal and smoothed with a Savitzky-Golay first-order filter with a 3-point window size.
Instantaneous swing velocity: the derivative of swing trajectory.

### Interlimb and whole-body coordination

Stance phase: relative timing of limb touchdowns to stride cycle of reference paw (FR). Calculated as: (stance time − stance time $_{reference\ paw}$) / stride duration.
Base of support: width between the two front and two hind paws during stance phase.

Body y displacement: y displacement of the body center during that stride.

Supports: support types were categorized by how many and which paws were on the ground, expressed as a percentage of the total stride duration for each stride. Paw support categories include 3-paw, 2-paw diagonal, 2-paw other/non-diagonal (homolateral and homologous), and 2-paw front (only) supports.

Double support for each limb is defined as the percentage of the stride cycle between the touch down of a reference paw to lift-off of the contralateral paw. Because at higher speeds (running), the opposing limb lifts off before the reference paw touches down, we included negative double support by looking backward in time, up to 25% of the stride cycle duration. Positive values of double support indicate that contralateral lift-off occurred after reference paw touch down, and negative values indicate that contralateral lift-off occurred before reference paw touch down. Note that front paw double support percentages include 2-paw front (only) support patterns as well as 3- and 4-paw support patterns in which both front paws were on the ground.

Tail and nose phases: For each speed bin, we correlate the stridewise tail and nose trajectories with the trajectory given by the difference between the forward position of the right paw and the forward position of the left paw (also normalized to the stride). The phase is then calculated by the delay in which this correlation is maximized.

Tail and nose peak-to-peak amplitude: the change between peak (highest amplitude value) and trough (lowest amplitude value) in y or z during a stride duration.

Variability: All variability analyses were based on coefficients of variation (CV).

## Principal component and linear discriminant analyses

The locomotor dataset consisted of a matrix of 45 features for each mouse and speed bin. Many gait features are highly correlated with speed (*Machado et al., 2015*), so to avoid inter-variable correlation and overfitting we first performed principal components analysis. The first 10 principal components explained more than 85% of the variance and the data projected onto these 10 principal components were used as input to the LDA (*Machado et al., 2020a*). LDA output is displayed separately for each speed bin to verify that the pattern of differences across groups was captured across all walking speeds.

## Statistical analysis

Electrophysiological data were analyzed using ANOVA and two-tailed t-tests. For GC input-output data, we performed a linear mixed effects analysis using *lme4* (*Bates et al., 2015*). As fixed effects, we entered current and genotype (with interaction term) into the model; cells were included as random term. P-values were obtained by likelihood ratio tests of the full model with the effect in question against the model without the effect in question. Statistical testing was performed in R (*R Development Core Team, 2017*). Effect sizes were calculated Cohen's d (for t-tests) or partial $\eta^2$ (for ANOVA) using the *effsize* and *sjstats* packages in R. Data in figures are presented as mean ± standard error of the mean (SEM). Eyeblink conditioning data were analyzed using t-tests. Locomotion data were analyzed in MATLAB with the Statistics toolbox. An independent samples t-test was used to test for differences in walking speed distributions (*Figure 7—figure supplement 1*). For all other gait parameters, analysis was performed on animal averages binned by speed using mixed effects models (*Bates et al., 2015*). Fixed-effects terms included speed and genotype; animals were included as random terms. *Supplementary file 1* reports effects of genotype for all locomotor measurements as F statistics from mixed ANOVAs with Satterthwaite degrees of freedom correction. No corrections were made for multiple comparisons.

## Acknowledgements

We thank Katharina Schmidt for genotyping and Hannah Monyer for providing the mice that originated our GluA4-KO colony. This work was supported by European Research Council Starting Grants #640093 (to MRC) and #679881 (to MM), by Portuguese Fundação para a Ciência e a Tecnologia PTDC/MED-NEU/30890/2017 (to MRC), by Swiss National Science Foundation Assistant Professor grant PP00P3_144816 (to MM) and Ambizione grant PZ00P3_174018 (to ID), and the Julius-Klaus Foundation (to ID).

## Additional information

### Competing interests

Megan R Carey: Reviewing editor, *eLife*. The other authors declare that no competing interests exist.

### Funding

| Funder | Grant reference number | Author |
| --- | --- | --- |
| European Research Council | 640093 | Megan R Carey |
| Fundação para a Ciência e a Tecnologia | PTDC/MED-NEU/30890/2017 | Megan R Carey |
| European Research Council | 679881 | Martin Müller |
| Swiss National Science Foundation | PP00P3_144816 | Martin Müller |
| Swiss National Science Foundation | PZ00P3_174018 | Igor Delvendahl |

The funders had no role in study design, data collection and interpretation, or the decision to submit the work for publication.

### Author contributions

Katarzyna Kita, Catarina Albergaria, Ana S Machado, Formal analysis, Investigation, Visualization, Writing - review and editing; Megan R Carey, Martin Müller, Conceptualization, Supervision, Funding acquisition, Writing - original draft, Writing - review and editing; Igor Delvendahl, Conceptualization, Formal analysis, Supervision, Funding acquisition, Investigation, Visualization, Writing - original draft, Writing - review and editing

### Author ORCIDs

Katarzyna Kita (iD) https://orcid.org/0000-0002-8371-7941
Catarina Albergaria (iD) http://orcid.org/0000-0001-8257-3600
Megan R Carey (iD) http://orcid.org/0000-0002-4499-1657
Martin Müller (iD) https://orcid.org/0000-0003-1624-6761
Igor Delvendahl (iD) https://orcid.org/0000-0002-6151-2363

### Ethics

Animal experimentation: This study was performed in strict accordance with national and institutional guidelines. All experiments were approved by the Cantonal Veterinary Office of Zurich (authorization no. ZH206/2016 and ZH009/2020) or by the Portuguese Direcção Geral de Veterinária (Ref. No. 0421/000/000/2015).

### Decision letter and Author response

Decision letter https://doi.org/10.7554/eLife.65152.sa1
Author response https://doi.org/10.7554/eLife.65152.sa2

## Additional files

### Supplementary files

- Supplementary file 1. Statistics for *Figure 7* and figure supplements.
- Supplementary file 2. Parameters used for the granule cell (GC) integrate-and-fire model.
- Supplementary file 3. Granule cell (GC) electrophysiological parameters.
- Transparent reporting form

## Data availability

All data generated or analyzed during this study are included in the manuscript and supporting files. Source data files have been provided for Figures 1–5; source code for simulations and modeling is available at https://github.com/delvendahl/GluA4_cerebellum_eLife copy archived at https://archive.softwareheritage.org/swh:1:rev:1d1c19853e18f8fbb3307eeecc64096e53d74820.

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
