## [Decision Letter]

**Acceptance summary:**

This work focusing on the cerebellum, a structure involved in the acquisition of arbitrary, complex motor reflexes, explores the cellular and behavioral effects of a genetically induced reduction of the expression of GluA4 (Gria4), an excitatory gluatamate neurotransmitter receptor. The authors show that synaptic transmission at the input layer to the cerebellar cortex is reduced, despite some compensation by other mechanisms, which are characterised. While overall locomotion is little affected, acquisition of a cerebellum-dependent conditioned eyeblink response is abolished. The authors try to link the cellular and behavioral phenomena via modelling of the cerebellar computation, although this is not definitive. The work is of high quality and of particular interest to cerebellar physiologists and neurocomputational modellers.

**Decision letter after peer review:**

Thank you for submitting your article "GluA4 enables associative memory formation by facilitating cerebellar expansion coding" for consideration by *eLife*. Your article has been reviewed by 2 peer reviewers, and the evaluation has been overseen by a Reviewing Editor and John Huguenard as the Senior Editor. The following individual involved in review of your submission has agreed to reveal their identity: Isabel Llano (Reviewer #1).

The reviewers found the slice experiments of high quality. However, the reviewers concluded that the work overall requires a more nuanced presentation and discussion. In addition, a few specific points about the experiments were also raised. These concerns are outlined below.

1. A first major concern regards the conclusion that learning and behavior have been constrained by the reduced coding expansion by the granule cell layer. The failure of learning may simply result from the inputs to Purkinje cells being too weak, too delayed or too unreliable to be an effective plasticity substrate for rapidly developing a conditioned response before the air puff. Although, to a large extent, the lower-level modifications will correlate with the higher-level coding expansion, it seems that concluding patterns cannot be separated because they produced no granule cell activity (to consider a logical extreme) and concluding that their separation is too difficult because of output similarity and saturation of learning are different. The conclusions should be moderated, and these issues should be raised in the discussion.

2. A second major concern revolves around the challenge of working with a systemic knockout. Logically, the behavioral effects on eyeblink conditioning could reflect interference with any part of the input-output loop. Furthermore, the major changes in NMDARs and tonic inhibition which impact the transfer function of the MF-GC synapses further complicate the interpretation of the results. In the absence of a granule-cell-specific knockout, the authors should moderate their conclusions and expand on these issues and the limitations of the systemic knockout approach in the discussion.

3. Along the same vein, the motivation for the MF-Go and PF-PC experiments should be clarified. Are those experiments intended to show that the cerebellar circuit is otherwise normal in the knockout or did the authors consider other scenarios in which the deletion of GluA4 could result in altered function at at those synapses (e.g. presynaptic AMPARs containing the GluA4 subunit)?

4. A third major concern is that, for the experiments in Figure 8, it is essential to show that the eyelid closure per se in not impaired. Figure 8B should include traces for KO as well as for WT mice, and there should be a separate panel for a comparison of the US response between WT and KO mice.

5. For Figure 2, statistics should be provided on the GC membrane resistance for the WT and KO group from recordings performed with high intracellular chloride as well as for those with low intracellular chloride. Furthermore, the Methods should describe better the current clamp protocols. What was the average Vm with no injected current and how do the GC firing vs injected current graphs for WT slices in control and in bicuculline (Figure 2C) compare to published results ? Furthermore, if the change described is the result of a change in tonic GABAAR activation, the use of "intrinsic" excitability seems confusing as the change is due to external tonic inhibition. This comment also applies to the Discussion.

6. Expanding the discussion on compensation (lines 398-409) would improve the manuscript. What are the possible causes for the reduction in tonic inhibition? Has this been observed in other KOs when AMPAR expression is reduced? As for the increase in NMDA, it may be possible to distinguish between pre- and post- synaptic mechanisms based on paired-pulse ratio protocols. Again, has such a change been previously observed in other mutants of AMPARs?

7. Potential differences pertaining to the two "compensatory" mechanisms between the slice and the in vivo situations should be discussed. What are the levels of tonic inhibition expected in behaving mice and how might they compare to the slice recordings reported here? Published work on how tonic inhibition affects GCs responses in vivo is available (eg. Duguid et al., 2012, J Neurosc. 32:11132-11143) and could help in these comparisons. Likewise, the possible contribution of NMDARs to synaptic currents/potentials in a slice vs in vivo should be considered given that the two situations differ in terms of the parameters that govern the activation of NMDARs.

8. Whether or not there are morphological changes in the GluA4-KO was not determined. Therefore, this possibility should be raised in the discussion. Does the decrease in AMPAR-dependent synaptic currents reflect a change in the number of AMPARs present at the postsynaptic density or is it due to a change in the function of a normal number of receptors? Are there any changes in the number of MFs that contact a GC?

9. Could regional differences in the expression of GluA4 contribute to the lack of effect on locomotion? Is the regional expression pattern for this subunit known? The discussion should explicitly note that slice electrophysiology was carried out in lobules III-IV while the two behavioral protocols tested include other cerebellar regions and should speculate as to how this may influence the interpretation of the functional role of the GluA4 subunit.

10. On page 3, line 90, the order for quantifications compared to the text is slightly confusing.

11. On page 5, line 107, the phrase "input-output" is slightly confusing because it is usually used to link injected current to spiking frequency. Perhaps "stimulus-response" would be clearer.

12. On page 6, line 130, the current factor in ANOVA should be treated as a repeated measure. It is a little odd to see p = 0.02 reported as "comparable".

13. On page 8, line 208 and line 215, The order for quantifications and text descriptions is confusing.

14. In Figure 6C, the small difference in sparseness does not seem to capture the full behavior, notably at the extremes in the heatmaps. How the median sparseness was calculated should be clarified.

15. More detail about how direct mossy fiber and indirect (via granule cell) inputs were connected in the perceptron should be provided. If it is all an exact reproduction of Gayco-Gajic et al., it may suffice to confirm this and maybe orient the reader within that reference.

---

## [Author Response]

The reviewers found the slice experiments of high quality. However, the reviewers concluded that the work overall requires a more nuanced presentation and discussion. In addition, a few specific points about the experiments were also raised. These concerns are outlined below.1. A first major concern regards the conclusion that learning and behavior have been constrained by the reduced coding expansion by the granule cell layer. The failure of learning may simply result from the inputs to Purkinje cells being too weak, too delayed or too unreliable to be an effective plasticity substrate for rapidly developing a conditioned response before the air puff. Although, to a large extent, the lower-level modifications will correlate with the higher-level coding expansion, it seems that concluding patterns cannot be separated because they produced no granule cell activity (to consider a logical extreme) and concluding that their separation is too difficult because of output similarity and saturation of learning are different. The conclusions should be moderated, and these issues should be raised in the discussion.

We agree with the reviewers that it is difficult to causally link the reduced expansion coding to learning and behavior of the GluA4-KO animals, and that behavior might as well be affected by a more basic impairment of the MF-GC-PC loop. We have addressed this issue by:

a) Analysing correlations between MF input and GC output in the feedforward-model. We examined the population correlation for GC and MF patterns according to Cayco-Gajic et al. 2017. Our data show that the raised threshold of GluA4-KO GCs increases the decorrelation achieved by the MF-GC network, albeit to a moderate extent (new Figure 6—figure supplement 1). The contribution of decorrelation to learning speed was comparable for both models, indicating that impaired learning is unlikely to be caused only by a strong reduction in GC activity. This is now described in the Results section:

“A decrease in the ratio of MF input strength to GC threshold might also reduce correlations in GC output patterns (Marr, 1969). […] Nevertheless, it is worth noting that a reduction in the fraction of active GCs per se might contribute to the reduced learning speed in the GluA4-KO model.”

b) Moderating our conclusions (see Discussion, first paragraph and Results, lines 316–319) and discussing alternative explanations for the failure of learning:

“The longer spike delay in GluA4-KO mice, together with the reduced MF→GC synaptic strength, raises the threshold of GC activation, leading to a loss of information (Billings et al., 2014; Cayco-Gajic et al., 2017), and potentially reducing the reliability of PC excitation (Galliano et al., 2013). Furthermore, the alterations of MF→GC transmission in the absence of GluA4 could cause delays and increase the temporal variability of PF-driven PC excitation in response to sensory stimulation, which may interfere with synaptic plasticity at PF→PC synapses (Suvrathan and Raymond, 2018).”

and

“Reduced expansion coding is expected to impair pattern separation and learning in downstream circuits. On the other hand, the effective rise of the GC threshold due to weak MF input strength may by itself decrease overlap between GC outputs, which would be predicted to improve learning speed (Marr, 1969)”

c) Changing the manuscript title to “GluA4 facilitates cerebellar expansion coding and enables associative memory formation”

2. A second major concern revolves around the challenge of working with a systemic knockout. Logically, the behavioral effects on eyeblink conditioning could reflect interference with any part of the input-output loop. Furthermore, the major changes in NMDARs and tonic inhibition which impact the transfer function of the MF-GC synapses further complicate the interpretation of the results. In the absence of a granule-cell-specific knockout, the authors should moderate their conclusions and expand on these issues and the limitations of the systemic knockout approach in the discussion.

The use of a systemic KO certainly represents a limitation of our study, as it is indeed conceivable that loss of GluA4 might interfere with learning outside of the circuits investigated in our slice recordings. However, we believe that a major contribution of GluA4 in the cerebellum is likely for the following reasons:

– GluA4 expression is generally low in the mammalian CNS outside of the cerebellum (see, e.g., Schwenk et al. 2014 doi: 10.1016/j.neuron.2014.08.044; Mouse protein atlas (Sjöstedt et al. 2020 doi: 10.1126/science.aay5947); Yamasaki et al. 2011 doi: 10.1523/JNEUROSCI.5601-10.2011).

– It is well established that delay eyeblink conditioning occurs within the cerebellum.

– Our experiments indicate that GluA4 predominantly contributes to MF-GC transmission in the cerebellum and MFs have been shown to convey the conditioning stimulus during eyeblink conditioning (Steinmetz et al. 1986 doi: 10.1037/0735-7044.100.6.878; Albergaria et al. 2018 doi: 10.1038/s41593-018-0129-x).

– Conditioned stimuli of different sensory modalities (visual and somatosensory) lead to similar deficits in associative learning (Figure 8C).

– Unconditioned responses to the air-puff stimulus are intact in the GluA4-KOs (new Figure 8B).

Nevertheless, we of course cannot rule out a contribution of GluA4 outside of the cerebellum and now raise these issues in the discussion in the following way:

- “We note that by using a systemic KO model, a functional role of the GluA4 AMPAR subunit at other synapses within the circuits responsible for the behavioral tasks assessed in our study may contribute to the observed results.”

– “Taking the limitations of a systemic KO model into account, our results indicate that GluA4-mediated MF→GC transmission is essential for cerebellar associative learning.”

3. Along the same vein, the motivation for the MF-Go and PF-PC experiments should be clarified. Are those experiments intended to show that the cerebellar circuit is otherwise normal in the knockout or did the authors consider other scenarios in which the deletion of GluA4 could result in altered function at at those synapses (e.g. presynaptic AMPARs containing the GluA4 subunit)?

We performed these experiments to address (i) if synapses onto Golgi cells depend on the GluA4 subunit, as the AMPAR expression in these neurons has not been fully characterized, and (ii) to study if the PF-PC synapse is altered in any way, e.g. by changes of GluA4-containing receptors in Bergmann glia that were shown to regulate PF-PC synaptic transmission. The motivation for these experiments is now stated more clearly in the Results section:

– “We next asked if GluA4 is involved in transmission at other synapses of the cerebellar input layer. Golgi cells (GoCs; Figure 1D) provide feedforward and feedback inhibition to GCs (Duguid et al., 2015), but it remains unknown if GluA4-containing AMPARs play a role in GoC excitation.”

– “The strong reduction of excitatory input onto GCs in GluA4-KO animals might alter GC output via their PFs. We therefore investigated PF inputs to GoCs, […]”

– “Whereas PCs do not express GluA4 (Lambolez et al., 1992), loss of this AMPAR subunit in Bergmann glial cells might impact PF→PC transmission (Saab et al., 2012). To examine PF→PC synapses […].”

4. A third major concern is that, for the experiments in Figure 8, it is essential to show that the eyelid closure per se in not impaired. Figure 8B should include traces for KO as well as for WT mice, and there should be a separate panel for a comparison of the US response between WT and KO mice.

We fully agree with the reviewers that intact eyelid closure is a prerequisite for interpreting the eyeblink conditioning results of the KO animals. Eyelid closure per se was not affected in KO mice. We have added two new panels to Figure 8C (formerly 8B) to show example traces of KO animals, which displayed a clear unconditioned response to the air-puff. As suggested by the reviewers we also added a separate panel to compare the US responses, which were not different between WT and KO (new Figure 8B).

5. For Figure 2, statistics should be provided on the GC membrane resistance for the WT and KO group from recordings performed with high intracellular chloride as well as for those with low intracellular chloride. Furthermore, the Methods should describe better the current clamp protocols. What was the average Vm with no injected current and how do the GC firing vs injected current graphs for WT slices in control and in bicuculline (Figure 2C) compare to published results ? Furthermore, if the change described is the result of a change in tonic GABAAR activation, the use of "intrinsic" excitability seems confusing as the change is due to external tonic inhibition. This comment also applies to the Discussion.

We are sorry for the lack of clarity in this section. We have revised the description of the experiments and results, and changed the analysis according to the suggestions of the reviewers. The changes are summarized in the following:

a) The data in Figure 2 originate from separate recordings with different intracellular solutions: We used a potassium-gluconate based solution for current-clamp recordings and a cesium-chloride based solution for voltage-clamp recordings of tonic conductance. We now explicitly mention these experimental differences in the legend of Figure 2: “GABAergic currents were isolated using 20 µM GYKI-53655 and a cesium-based intracellular solution with high intracellular Cl− to maximize GABAA receptor mediated currents (see Methods).” In addition, we have added data on GC membrane resistance, membrane capacitance and series resistance for these recordings, as suggested (new Supplementary Table 3).

b) We now describe the current-clamp protocol – including the average Vm without current injection – in the Methods section as follows: “GC excitability was assessed in current-clamp mode by tonic current injections (duration, 200 ms). For each sweep, the current amplitude was incremented by 5 pA, until a maximum of 40 pA. Current injections were applied from the GC resting membrane potential, which was not different between KO and WT (−97.1 ± 0.6 mV, n = 37 vs. −97.8 ± 0.6 mV, n = 46; p = 0.46).”

c) The frequency-current relation that we observed in WT granule cells is well within the range reported previously (~2–6.5 Hz/pA, with the majority of studies around 4–5 Hz/pA). We now state this in the Methods section:

“The average gain in WT under control condition was 4.3 ± 0.4 Hz (n = 36 GCs), in line with previous studies (Gall et al., 2003; Rizwan et al., 2016; Rudolph et al., 2020; Soda et al., 2019; Straub et al., 2020).”

d) The effect of blocking inhibition by bicuculline on the frequency-current relationship is also consistent with previous work. We now state this in the Results section: “Blocking inhibition increased the gain and reduced rheobase in WT (Figure 2D), consistent with previous studies (Hamann et al., 2002; Rothman et al., 2009; Rudolph et al., 2020).”

e) Regarding the terminology, we fully agree that the use of “intrinsic excitability” might be misleading. We have therefore changed the wording throughout the manuscript, referring to “input-output relationship” instead (see also response to point #11).

6. Expanding the discussion on compensation (lines 398-409) would improve the manuscript. What are the possible causes for the reduction in tonic inhibition? Has this been observed in other KOs when AMPAR expression is reduced? As for the increase in NMDA, it may be possible to distinguish between pre- and post- synaptic mechanisms based on paired-pulse ratio protocols. Again, has such a change been previously observed in other mutants of AMPARs?

We have now extended the discussion on the compensations we observed in GluA4-KO GCs and on their potential implications. Intriguingly, there is some evidence for compensatory changes in inhibition and NMDAR-mediated transmission in mouse models with impaired AMPAR expression:

– Interestingly, *stargazer* and *waggler* mice that lack functional AMPARs in cerebellar GCs – and thus have a strong impairment of MF→GC transmission – show a reduction of alpha6 GABAR levels and a concomitant reduction in tonic inhibition (Payne et al. 2007 doi: 10.1074/jbc.M700111200; Chen et al. 1999 doi: 10.1073/pnas.96.21.12132). Together, these findings suggest that changes in tonic inhibition could be a general compensatory mechanism in GCs upon strong impairment of synaptic excitation.

– GluA1-KO mice show an increase in glutamate and NMDAR levels in the hippocampus (Chourbaji et al. 2008 doi: 10.1096/fj.08-106450).

– In the reticular thalamic nucleus of *stargazer* mice, NMDAR conductance is increased while AMPAR conductance is reduced (Lacey et al. 2012 doi: 10.1523/JNEUROSCI.5604-11.2012).

– On the other hand, several studies using AMPAR-KO mice did not observe changes in NMDAR-mediated transmission.

These studies are now referred to in the revised manuscript.

We have also addressed potential causes for the observed changes in tonic inhibition and NMDAR-EPSCs:

– First, we performed western blot analysis to examine levels of the alpha6 GABAR subunit that mediates tonic inhibition of cerebellar GCs (Brickley et al. 2001 doi: 10.1038/35051086). Our data show a slight but consistent reduction of alpha6 GABAR levels in KO mice (updated Figure 2—figure supplement 1).

– We have performed paired-pulse recordings of NMDAR-EPSCs, which were unaltered (new Figure 5—figure supplement 1). To gain further insights into pre- or postsynaptic mechanisms, we next applied statistical moments analysis of quanta (Holler et al. 2021 doi: 10.1038/s41586-020-03134-2) to our NMDAR-mediated EPSCs. This analysis revealed an increase in the number of release sites without apparent changes in release probability or quantal size (new Figure 5—figure supplement 1). A comparable change in release site number was also observed when analysing AMPAR-meditated EPSCs. Together, these new data indicate that an increase in presynaptic release sites causes the larger NMDAR-EPSCs in GluA4-KO GCs, consistent with our previous results from presynaptic recordings in these animals (Delvendahl et al. 2019 doi: 10.1073/pnas.1909675116).

These additional data provide more insights into the mechanisms of compensation in GluA4-KO GCs.

We have expanded the discussion on compensations as follows (see also our answer to #7):

“Intriguingly, we observed lower α6 GABAAR levels by Western blot analysis in the GluA4-KO cerebellum, consistent with previous results from stargazer and waggler mice (Chen et al., 1999; Payne et al., 2007). These findings suggest that reduced α6 GABAAR expression contributes to the modulation of GC tonic inhibition upon loss of AMPARs.”

and

“The larger NMDAR-EPSCs in GluA4-KO are most likely caused by enhanced glutamate release (Chourbaji et al., 2008; Delvendahl et al., 2019) that is driven by an increase in the number of presynaptic release sites, but we cannot entirely exclude postsynaptic contributions.”

7. Potential differences pertaining to the two "compensatory" mechanisms between the slice and the in vivo situations should be discussed. What are the levels of tonic inhibition expected in behaving mice and how might they compare to the slice recordings reported here? Published work on how tonic inhibition affects GCs responses in vivo is available (eg. Duguid et al., 2012, J Neurosc. 32:11132-11143) and could help in these comparisons. Likewise, the possible contribution of NMDARs to synaptic currents/potentials in a slice vs in vivo should be considered given that the two situations differ in terms of the parameters that govern the activation of NMDARs.

In the revised manuscript, we now specifically discuss the changes in inhibition and NMDAR-mediated transmission as well as their potential implications in an in vivo situation (see also response to point #6):

– “Our slice data of GC tonic inhibitory conductance are similar to observations in vivo (Duguid et al., 2012), where tonic inhibition controls GC excitability (Chadderton et al., 2004) and EPSC-spike coupling (Duguid et al., 2012). The reduced inhibition in GluA4-KO mice may thus enhance sensory information transfer at the cerebellar input layer.”

– “Compared with slice recordings, the contribution of NMDARs to GC synaptic excitation is likely to be larger in vivo (Zhang et al., 2020), due to the more depolarized membrane potential and increased spontaneous input in GCs.”

8. Whether or not there are morphological changes in the GluA4-KO was not determined. Therefore, this possibility should be raised in the discussion. Does the decrease in AMPAR-dependent synaptic currents reflect a change in the number of AMPARs present at the postsynaptic density or is it due to a change in the function of a normal number of receptors? Are there any changes in the number of MFs that contact a GC?

We did not observe obvious differences in cerebellar gross anatomy between WT and GluA4-KO mice (new Figure 1—figure supplement 3). However, it is conceivable that the KO animals develop altered synaptic ultrastructure or MF-GC connectivity ratio, with the latter likely being limited by morphological constraints (Gilmer and Person 2017 doi: 10.1523/JNEUROSCI.0588-17.2017).

Regarding AMPARs in GluA4-KO GCs, our data suggests that there is no compensatory expression of other high-conductance AMPAR subunits. This makes it likely that the remaining AMPARs are GluA2 homomers, which have a low conductance (Swanson et al. 1997 doi: 10.1523/JNEUROSCI.17-01-00058.1997). Whether the amount of AMPARs is altered remains to be determined.

We now discuss the possibility of morphological changes in the revised manuscript:

“Additional compensations could occur at the level of morphology. While we did not observe obvious differences in cerebellar gross anatomy between WT and GluA4-KO mice, changes in, for instance, MF bouton ultrastructure or the number of MFs contacting a GC could affect the properties of GCs and MF→GC synapses in GluA4-KO mice.”

9. Could regional differences in the expression of GluA4 contribute to the lack of effect on locomotion? Is the regional expression pattern for this subunit known? The discussion should explicitly note that slice electrophysiology was carried out in lobules III-IV while the two behavioral protocols tested include other cerebellar regions and should speculate as to how this may influence the interpretation of the functional role of the GluA4 subunit.

First, we would like to note that there was a typo in the Results section, resulting in a discrepancy between Methods and Results. Our recordings were indeed performed in lobules III–VI, which has been corrected in the Results section of the revised manuscript. These lobules are involved in both eyeblink conditioning and locomotion (e.g., Giovannucci et al. 2017 doi: 10.1038/nn.4531; Markwalter et al. 2019 doi: 10.1523/JNEUROSCI.0086-19.2019; Muzzu et al. 2018 doi: 10.1371/journal.pone.0203900). Second, in situ hybridization data for the GluA4 subunit show a homogeneous pattern throughout the cerebellum (Allen Brain Atlas; Yamasaki et al. 2011 doi: 10.1523/JNEUROSCI.5601-10.2011), making it likely that cerebellar GCs rely on this AMPAR subunit in general. However, we cannot exclude that GCs in certain cerebellar regions are affected to a lesser extent by GluA4-KO and that this could play a role in the behavioural results. This issue is addressed in the discussion of the revised manuscript:

“Our slice recordings were confined to cerebellar lobules III–VI and regional differences in GluA4 expression could in principle contribute to the preserved locomotion in GluA4-KO mice. […] We therefore consider it unlikely that regional differences play a major role in the differential effects on eyeblink conditioning vs. locomotion in GluA4-KO animals.”

10. On page 3, line 90, the order for quantifications compared to the text is slightly confusing.

The order of numerical values has been changed and now consistently states KO data before WT to avoid confusion.

11. On page 5, line 107, the phrase "input-output" is slightly confusing because it is usually used to link injected current to spiking frequency. Perhaps "stimulus-response" would be clearer.

We have changed the wording accordingly.

12. On page 6, line 130, the current factor in ANOVA should be treated as a repeated measure. It is a little odd to see p = 0.02 reported as "comparable".

We would like to thank the reviewers for highlighting this issue and apologize for the confusion in the description of the results. In the revised manuscript, we have now included an analysis using a linear mixed-effects model with current, drug, and genotype as fixed factors and individual cells as random variable. This analysis treats current as a repeated measure as suggested. The description of these results has also been revised for clarity.

13. On page 8, line 208 and line 215, The order for quantifications and text descriptions is confusing.

The order of these quantifications has been changed.

14. In Figure 6C, the small difference in sparseness does not seem to capture the full behavior, notably at the extremes in the heatmaps. How the median sparseness was calculated should be clarified.

The median was calculated across different levels of active mossy fibres for any given correlation radius (i.e. by row in the heatmaps). We now describe how the median was calculated in the figure legend: “Median is calculated across fractions of active MFs.” (Figure 6 legend). To improve the clarity of the figure, we have also rotated the heatmaps to show correlation radius on the abscissa.

15. More detail about how direct mossy fiber and indirect (via granule cell) inputs were connected in the perceptron should be provided. If it is all an exact reproduction of Gayco-Gajic et al., it may suffice to confirm this and maybe orient the reader within that reference.

Indeed, the connection of inputs was performed as in Cayco-Gajic et al. 2017. This is now mentioned in the Methods section:

“Input patterns were connected to the perceptron as described in Cayco-Gajic et al. (2017).”